# 'The world is just so fast, and I'm not fast… it's just really, really difficult to keep up': A qualitative exploration of the lived experience of adults with Developmental Coordination Disorder

**Rebecca Murray**[1]*, **Cara E. Staniforth**[1,2], **Lucy H. Eddy**[3]

**1** Department of Psychology, University of Bradford, Bradford, United Kingdom, **2** Centre for Applied Education Research, Bradford, United Kingdom, **3** School of Psychology, Northumbria University, Newcastle, United Kingdom

* r.murray2@bradford.ac.uk

## Abstract

### Background

Developmental Coordination Disorder (DCD) is an under-recognised neurodevelopmental condition impacting 5–6% of the population. There is a growing evidence base showing the deleterious impacts of poor motor skill development, however there is a distinct lack of research gathering in-depth insights which explore the impact of DCD within both childhood and adulthood.

### Methods

Ten online lived experience interviews were undertaken with adults who have a diagnosis of DCD/ Dyspraxia (8 females, 2 males), lasting between 30 minutes and one hour. Participants were recruited both nationally and internationally. Lived experience interviews focused on experiences in primary care, education, friendships, wellbeing, employment and romantic relationships. Data were transcribed and analysed using inductive thematic analysis.

### Results

Emergent themes highlighted a major lack of societal awareness in all life domains, which often led to participants facing difficulties navigating health, education and workplace systems for support, resulting in relative abandonment and a lack of validation within their lived experience of DCD.

### Conclusion

Individuals with DCD often report feeling alone, isolated and misunderstood in a world which presents challenges for them across all aspects of life. There is an

**Data availability statement:** Due to the sensitive nature of lived experience interviews, the research team believe that it would be inappropriate to make the data available. However, if anyone requests the data, we will provide heavily redacted transcripts - fully anonymised to ensure participant anonymity. Ethical approval from the University of Bradford does not allow the data from this study to be shared without restriction. To access the data in an anonymised and redacted format, individuals can contact the ethics committee (ethics@bradford.ac.uk).

**Funding:** The work of the senior author (L.H. Eddy) and co-author (C.E. Staniforth) was supported by a grant from the Waterloo Foundation (ref: 27665413). There was no additional external funding received for this study. The funders had no role in study design, data collection and analysis, decision to publish, or preparation of the manuscript.

**Competing interests:** The authors have declared that no competing interests exist.

urgent need for the profile of DCD to be raised by insider voices as for many, DCD often transcends diagnostic criteria to wider challenges, for example executive functioning. Fundamentally, more needs to be done to ensure a lifespan approach to DCD, to allow greater opportunities for adults with a diagnosis to thrive alongside their 'neurotypical' peers.

## Introduction

Developmental Coordination Disorder (DCD) is a neurodevelopmental condition (NDC) in which motor skill challenges that begin in early childhood have a significant and persistent impact on everyday activities, such as dressing, cooking, and engaging with education and play [1–5]. Although DCD is the formal diagnosis, it is also known as and referred to as Dyspraxia which reflects the history of the condition. DCD is often, although not exclusively diagnosed in childhood, however DSM-5 diagnostic criteria [4] and European Academy for Childhood Disability's (EACD) international recommendations [6] emphasise the life-long impacts of the condition and stipulate that challenges persist into adulthood.

Between 30–70% of children that are diagnosed with DCD report difficulties that persist into adulthood [7,8]. However, there are far fewer studies exploring implications for later life. Of the literature that does exist there is evidence to suggest that young adults with DCD had some continuing difficulties with motor skills, which were not easily overcome, in addition to limited strategies to support the complexities of their challenges in daily living [9]. This has been shown to impact on the continued education of these individuals, who, for example continue to struggle with handwriting and fine motor activities which are central to university life and beyond [8,10]. In addition, there are reports of challenges with skills which facilitate independence, such as learning to drive [11,12], with one study highlighting that these challenges extended beyond the motor skills needed to perform the act of driving, to challenges with navigating and using maps [10].

Another large part of transitioning to independence in early adulthood is moving away from home. However, studies have found that as many as 50% of adults with DCD find self-care activities challenging [9], with tasks such as cooking, and cleaning leaving adults with DCD enduring the cumulative burden of consequential fatigue which plays a significant role in their day to day lives [10]. Previous literature has suggested that such challenges with self-care skills could reflect associated difficulties with executive functioning skills, such as planning, processing speed, organisation and prioritisation [9,12]. As many as 30% of adults highlight executive functioning as an area of concern when self-referring for a DCD assessment [13]. It is, however, important to note that DCD is often diagnosed with one or more co-occurring condition(s), such as Autism (50%) and ADHD (30%) [14].

Although the diagnostic criteria focus solely on motor skill challenges, there is evidence of secondary impacts of DCD for mental health and wellbeing [1,15]. For example, some adults with DCD have reported being bullied [16], experiencing a

lower quality of life [17], and/or heightened anxiety levels related to other people's perceptions of their abilities, significant enough to prompt access to counselling services [10]. Literature shows that these challenges with mental health and wellbeing can lead to social isolation amongst adults with DCD [9,12]. For example, with some choosing to withdraw from and/or avoid situations which highlight their difficulties [13,16,18]. This avoidance has been evidenced by low levels of participation in physical activity for adults with DCD [12], and an increased likelihood of living at home with parents whilst at university [7].

Another key milestone in the transition to adulthood independence is gaining employment [19]. Previous research has highlighted parental concerns about their child's ability to adapt to or engage with workplace settings [9]. Young adults with DCD also share these concerns as they have reported challenges with finding and maintaining employment to be self-sufficient [20]. Given that employment has also been found to play a significant protective role in mental health for adults with DCD, including reduced likelihood of depression, and increased quality of life ratings [20], such challenges could have wider implications for the DCD population. With legislation on the horizon pertaining to increasing the proportion of the workforce in the UK that have disabilities [21], including NDCs, it is crucial that more research is done to see how we can best support adults with DCD in workplaces.

To date, a lot of research on DCD in adulthood has comprised questionnaire data [9,13,16,17,20], which can limit the depth and breadth of the data obtained [22]. Of the few studies which take a more in-depth qualitative approach [11], these have a narrow focus on either on one period of adulthood, such as university life [8,10], or one specific challenge faced by those with DCD for example, driving [11], or daily participation [23]. There is therefore a lack of qualitative research exploring the experiences of adults with DCD holistically across life domains to understand the intersectionality of challenges. This is particularly the case in relation to the secondary impacts of DCD including friendships, romantic relationships and employment. Similarly, there is a paucity of in depth accounts of retrospective experiences and interactions within health and education from the adult perspective, with current literature relying mainly on parental perspectives [24–26]. This study therefore aimed to use semi-structured interviews with adults with DCD to understand challenges faced throughout childhood, and all domains of adulthood in order to respond to current gaps in the literature and give voice to an underheard population.

## Methods

### Design

This research utilised a qualitative interview design, which facilitated exploration of lived experience. This enabled a holistic picture of the challenges faced by adults with DCD and how these challenges change throughout the life course. To ensure a flexible approach that encouraged sharing of experience whilst also maintaining relevance for research aims, semi-structured interviews were utilised [27].

### Participants

Participants were recruited via purposive sampling, via adverts on social media (Facebook, Twitter, BlueSky, LinkedIn and Instagram) and through contacting four English-speaking national DCD charities in Europe, Asia, and North America. Inclusion criteria required participants to have a formal diagnosis of DCD or Dyspraxia (regardless of age diagnosed), and English fluency proficient enough to participate in interviews.

Ten adults with a diagnosis of DCD/ Dyspraxia participated in interviews. This is deemed an appropriate sample size for lived experience of this nature [28]. Furthermore, the ten interviews presented consistent narratives which reflected data saturation [28] and thus no further recruitment was required. Those with a Dyspraxia diagnosis were included as this term was previously used to diagnose, and diagnosis in adulthood can still be classed as Dyspraxia [6]. Nine participants lived in the United Kingdom (UK), and one lived in New Zealand. Participants were diagnosed between the ages of 5 and 49 years.

The sample comprised 8 females and 2 males and 5 had a co-occurrence (two with Autism, two with Dyslexia, and one with ADHD). Given the high co-occurrence rates within academic literature, this was to be expected [14,29]. All diagnoses were self-reported and not independently verified by researchers. Participant age was not collected, and therefore age-related analyses or generational interpretations are limited. Participants were recruited from 6th January 2025–4th March 2025. Data saturation was achieved via the ten participants that were recruited, and therefore no further recruitment was required.

## Materials

Interview questions were generated to gain insights into challenges faced within both childhood and adulthood, including accessing diagnosis and aftercare within healthcare services as well as experiences related to family, friendships, education, employment and romantic relationships (See Table 1).The topic guide was intentionally broad to capture insights related to the challenges faced by children and adults with DCD according to previous literature [20,30–32], whilst also enabling a more holistic lens in order to explore underheard challenges. These questions were not pilot tested.

## Procedure

Interested participants contacted the research team via email addresses provided on adverts. Researchers answered any questions and sent through an information sheet detailing the purpose of the study. Participants that wanted to take part were asked to complete a written consent form and return this via email. Once consent had been obtained, a time and date were set for the online interview (via Microsoft Teams) or in person (depending on personal preference) at the convenience of participants. All authors were involved in interviews. One participant requested an in-person interview, the rest were completed online. When participants requested questions in advance, these were sent to ensure they felt as comfortable and prepared as possible. A minimum of two authors were present at all interviews to ensure all relevant questions were asked, and areas for discussion were fully covered. No one outside of the research team attended the interviews. Participants were asked seven questions, with prompts for further discussion used where necessary (see Table 1). At the end of the set questions, participants were asked if there was anything else they would like to discuss in relation to their DCD diagnosis, to ensure their experiences were fully presented. Interviews lasted between 30 minutes to an hour.

All interviews were recorded and transcribed. After the interviews, Microsoft Teams transcripts were downloaded, and the authorship team checked content against recorded content, to ensure transcripts fully reflected conversations. The in-person interview was recorded via tablets and later transcribed by the research team. Participants were given pseudonyms, and any potentially identifiable information was removed to ensure anonymity. After anonymisation, transcripts were sent to participants for member checking to ensure they were reflective of the participants' true feelings, thus improving the credibility and trustworthiness of the data [33]. Participants were given two weeks to review and amend transcripts, before the data were analysed. Ethical approval for this study was granted by the University of Bradford Ethics Committee (reference: E1026).

## Analysis

Within an interpretivist framework, thematic analysis was utilised to analyse interviews using the six stages of Braun and Clarke (2006; [34]): (i) Familiarisation with the data: Interview data were transcribed, and each interview transcript was read and re-read by all authors. This allowed those initial ideas to be explored independently; (ii) Generation of initial codes: Using NVivo, all authors systematically coded any defining features within and across transcripts; (iii) Search for themes: Codes were organised into themes and data was subsequently organised to evidence themes; (iv) Review themes: The reviewing of themes allowed confirmation that codes were appropriate and sufficiently substantiated; (v) Define and name themes: All authors refined the specifics of each theme; (vi) Produce the report: The quotes included in the results section were selected based on relevance to research aims, and their vibrant and compelling nature, which effectively articulated the lived experiences which transcended participants' data.

**Table 1. Interview Questions.**

| Question | Prompts |
| --- | --- |
| Can you tell us about the process you went through to get a DCD/ Dyspraxia diagnosis? | Did you encounter any barriers?<br>Were there any helpful facilitators?<br>Age of diagnosis? |
| Did you feel sufficiently supported by your family (immediate and wider) after your diagnosis? | Did you come up against any scepticism or doubt? (If yes, can you tell me about this?)<br>Were you confronted by any non-believers? (If yes, how did you cope with this?)<br>Did you feel supported at home? (Can you describe why?) |
| Did your diagnosis impact on your experience in education (including school, FE and HE) in relation to peer and teacher perceptions and support? | Were there any positive impacts?<br>Were there any negative impacts?<br>Were there any particular motor skills which had the greatest impact on your experience? |
| Has your diagnosis influenced your job/career? | Has your diagnosis impacted on your choice of job/career?<br>Have you always disclosed your diagnosis when applying for jobs?<br>Have you faced any barriers and/or facilitators in your workplace (has there been effective support)?<br>Were there any particular motor skills which had the greatest impact on your experience? |
| Has your diagnosis influenced your friendships? | Have you felt supported within these friendships?<br>Have you always shared your diagnosis with friends?<br>Has your diagnosis ever been considered a problem in your friendships?<br>Were there any particular motor skills which had the greatest impact on your experience?<br>Beyond friends and family did you reach out for support elsewhere (e.g., support groups) – can you tell me why? |
| Has your diagnosis influenced your relationships (romantically)? | Have you felt supported within these relationships?<br>Have you always shared your diagnosis with partners?<br>Has your diagnosis ever been considered a problem in your relationship?<br>Were there any particular motor skills which had the greatest impact on your experience? |
| Do you feel understood in relation to your diagnosis? | |
| Is there anything you would like to tell us about your experiences that we haven't covered? | |

All authors were present when each transcript was analysed, which allowed consensus to be reached when disagreements and contrasting interpretations were tabled.

## Research team and reflexivity

The lead female author (RM; PhD) is a Lecturer in Psychology with expertise in qualitative methodology and underheard populations. CES (MSc) is a Peer Research Associate – her expertise is in motor development. AS (BSc) was an undergraduate student at the time of data collection, working on this project for her dissertation. LHE (PhD) was an Assistant Professor in Psychology at the time of data collection, with expertise in childhood motor development and DCD. The

research team's interests and expertise facilitated the current project; however, participants experience of DCD was central to analysis, and thus any potential for bias was mitigated by all authors keeping a reflexive journal which allowed decision making processes to be transparent within the team.

Prior to consenting, participants were sent an information sheet which detailed the reasons for conducting the research. At the start of the interviews, authors introduced themselves and explained their role in the project and their motivation for the topic. No formal rapport was built prior to the interviews; however participants were given contact details to liaise with the team to answer any questions they had beforehand.

## Results

The current analysis presents two major themes and 19 subthemes through which participants shared their experience across all life domains (See Table 2).

### Theme 1: Impact of DCD

The analysis revealed a major theme of the impact of DCD which transcended childhood into adulthood. Moreover, it became evident that DCD not only impacted on motor development (the cornerstone of diagnostic criteria) but also played a role in developmental milestones (e.g., driving), cognitive load, employment opportunities, friendships and romantic relationships as well as mental health.

**Subtheme 1.1: Motor skill challenges in childhood.** Many participants initially reflected upon challenges they faced in relation to motor skills in childhood, including practices which are often deemed key milestones, such as learning to ride

**Table 2. Table of Themes.**

| Major Theme | Subtheme(s) |
| --- | --- |
| 1. The impact of DCD | 1.1 Motor skill challenges in childhood |
| | 1.2 Acquiring the label of clumsy |
| | 1.3 Developmental milestones which enable independence |
| | 1.4 Cognitive load |
| | 1.5 Challenging peer relationships in childhood |
| | 1.6 Commonality in difference in adult friendships |
| | 1.7 Commonality in difference in romantic relationships |
| | 1.8 Lack of equity in employment opportunities |
| | 1.9 Mental health and wellbeing across the life course |
| | 1.10 The positive role of diagnosis |
| 2. Lack of societal awareness, understanding and support | 2.1 Non-believers in education |
| | 2.2 Experiences of discrimination within the education system |
| | 2.3 Lack of resources and support within healthcare |
| | 2.4 Under-supported and undervalued at work |
| | 2.5 Misconceptions and ignorance within families |
| | 2.6 Abilities seen as not conducive to a DCD diagnosis |
| | 2.7 Denied validation |
| | 2.8 Inherent lack of awareness compared to other neuro-developmental conditions |
| | 2.9 Considerable gap in community level support and advocacy |

a bike. This was particularly challenging for participants, who reported various barriers in relation to their abilities and the expectations of others.

*It took me a week to learn to ride a bike… And I liked to ride my bike around the square and consistently every single circuit I threw myself into the same set of bins that didn't move. I just didn't have the depth perception...* – Blake (Female, no co-occurring NDCs)

*With riding a bike and stuff, my dad, he put so much pressure on me because he wanted me to do it, and I understand now he just wanted me to do it. But at the time, I got really upset about it because obviously, I'm a 5-year-old child, I can't ride a bike. Well, I can now, but it took so long. Like my brother, he just got on it and he could go* – Ella (Female, co-occurring Dyslexia)

*And I'm not sure if that's because I was a girl because both my brothers engaged quite a bit in sport and stuff. And I was just allowed to get away with it… I didn't really learn to ride a bike… All of this stuff I've done later on in life with a lot of effort, and I'm not very good at that stuff at all. But yeah, it's just like, why was I allowed to get away with that? I often think about that.* – Izzy (Female, no co-occurring NDCs)

Despite Ella having co-occurring Dyslexia, the challenges she faced whilst learning to ride a bike were not related to this diagnosis as they were procedural and directly aligned with DCD diagnostic criteria in the DSM-5 which states that motor skills are below age expected levels [4]. For many, the motor skill of riding a bike was not the only challenge, rather there was an additional burden associated with their observations of their typically developing siblings who physically thrived.

**Subtheme 1.2: Acquiring the label of clumsy.** Participants also reflected on their childhood experiences of physical activity, in relation to both structured sessions (such as Physical Education) and free play. For many participants, this was not a positive experience, as they reported acquiring the label of 'clumsy', which undermined the challenges they faced.

*I just thought I was rubbish at any kind of physical activity because I avoided it like the plague... I didn't have any confidence with physical activity. I did nothing. I would not do anything. I hated PE. I wouldn't do any extracurricular clubs and stuff. And it's really just interesting to me, like I am very clumsy. I'm not very good.* – Izzy (Female, no co-occurring NDCs)

*I am extremely clumsy. I'm also I'm quite tall as well, so I think my clumsiness is really obvious. I was terrible at PE, like absolutely terrible… they [school] would read out the results at the end of the year and I'd always be bottom. Sports day was awful, like it was really obvious I had issues of some sort.* – Katherine (Female, co-occurring Dyslexia)

*I was about three years old, and I was running around outside and I was wearing a bobble hat as it was winter and the hat fell over my eyes because it was too big for me... My parents were moving [house], so they weren't necessarily paying attention to me. And instead of stopping running… I just kept running and I ran into a brick wall…I didn't stop running…I hit my head on that wall so many times… by the age of 10 I'd been admitted to hospital something like 15 times with serious injuries, three of which were broken bones, and then a lot of them were concussions and things like that from like falling down the stairs or falling off climbing frames or whatever. And it got so serious at one point that there was. My parents were basically approached and kind of threatened with 'if she's here again, we're going to call social services'* – Blake (Female, no co-occurring NDCs)

Whilst Katherine had a co-occurring diagnosis of Dyslexia, the clumsiness she describes above cannot be attributed to typical Dyslexia presentation, whereas DCD, previously known as 'clumsy child' describes her experiences well. Whilst 'clumsiness' is an outdated term associated with DCD, some participants drew upon the label 'clumsy' on reflecting upon their

experiences in childhood. However, the reality of their 'clumsiness' sometimes had in direct implications for their health and wellbeing, for example in Blake's case, multiple visits to emergency care resulted in scrutiny for their family by social services.

**Subtheme 1.3: Developmental milestones which enable independence.** Participants subsequently highlighted that challenges with motor skills persisted into adulthood, which is often missing from the wider narrative around DCD. Typical developmental milestones which enable independence, such as learning to drive were rehearsed as problematic for participants. For many, passing their driving test took multiple attempts, and even after passing, participants noted how exhausting the practice of driving was.

*I passed my driving test after 10 attempts in a manual car and after four years of lessons* – Liam (Male, co-occurring ADHD)

*I've started driving again recently, but I've really hated driving. And people say… "why aren't you driving?", and if I say, "well, you know, I'm Dyspraxic, I really struggle with spatial awareness'… After I'd passed my test… I really actually didn't want to drive. I didn't feel confident about it… But the thing that I'm really, really struggling with and has caused me a number of panic attacks is parking. I just really, really struggle with spatial awareness* – Amy (Female, no co-occurring NDCs)

*So I learnt to drive when I was 17 which… I didn't really know how to drive till I was about 25, when suddenly I went 'Oh my God, that's what they mean by finding the bite on the clutch'… I have progressively I just gotten worse. I just get so stressed, I think it's something about modern cars lots going on…I can't get anywhere without using SAT NAV. I have to have the voice on the SAT NAV to be able to follow it…I can't read maps. I can't do any of that.* – Izzy (Female, no co-occurring NDCs)

One participant (Liam) had co-occurring ADHD, and therefore the challenges he faced upon learning to drive may also in part be attributed to the attention required to pass a driving test. However, other participants with sole diagnosis of DCD also shared similar experiences not only upon learning to drive, but also the practice of driving more broadly. Thus, whilst driving is largely a motor task, participants also attributed some of these challenges to the cognitive demands associated with navigation and spatial awareness etc.

**Subtheme 1.4: Cognitive load.** On unpicking the impact of DCD on their everyday lives, participants reported struggling with processing, planning and organisation not just when driving but also in daily life more broadly. Participants revealed the reality of life within which taken for granted practices demanded significant efforts which could arguably reflect increased demands on cognitive load.

*I don't think even I really understood...I knew that I struggled with processing things and organising and understanding things sometimes. But I think...I don't think it really hit me how much it affects me in my wider life until I was older and had left home and I was struggling being at work, a self-employed working mum with a husband and children, you know, that I thought actually I really, really struggle just with keeping up with everything, you know, the basic things.* – Amy (Female, no co-occurring NDCs)

*If I am not concentrating things like my spatial awareness, orientation not only to physical space, but time like I'm so bad with time management and just organising myself, things like putting clothes away…if I'm not switched on, I just can't remember what to do or I'll do things twice or something... I'm hopeless. I joke that I could get lost in my own house, but I genuinely could* – Izzy (Female, no co-occurring NDCs)

*My time management can be really, really bad. I'm either really early or late… my planning is shocking…I thought, I better look through my emails and then found out I'm meeting you today [laughs], and I thought, 'oh I've forgotten about that one'* – Samuel (Male, co-occurring Autism)

 

Whilst cognition is not mentioned within diagnostic criteria (aside from exclusionary criteria related to intellectual deficits), participants reported that challenges related to memory, information processing, organising, time keeping and concentration all played a role in their lived experience of DCD. It is worth noting that one participant had co-occurring Autism (Samuel) which may have played a role in his experiences related to cognition, however participants with a DCD diagnosis alone, also reported such challenges.

Additionally, participant reported cognitive challenges related to their lived experience of DCD, sometimes left them feeling exhausted, even after 'simple' tasks.

*The tiredness as well because like to do stuff like even just simple… it just takes longer… even making the bed can be just like, no… when I was at home, they'd be like, 'why are you having a nap during the day', like, 'you should be up doing stuff'. And even now if I have a nap during the day, I'm like 'oh, maybe I should be doing stuff', like around the house.* – Ella (Female, co-occurring Dyslexia)

*I can do anything, but I've absolutely got to concentrate like hell. Even just getting like driving for example, you know if I'm going somewhere new. It's not that I can't do it, but I will have to concentrate so hard that it's absolutely exhausting. And if I forget to concentrate, I literally could be anywhere. I've done that where I've set off with the kids in the car to take them swimming or something and I've ended up at my parent's house and they're like 'what are you doing?' Because I just forgot to concentrate… what I've realised is that I am always having to stay completely switched on in day-to-day life and if I'm not, I do make mistakes and I do have accidents. And it's that's quite exhausting, actually* – Izzy (Female, no co-occurring NDCs)

Some participants revealed that exhaustion was a hidden burden associated with their lived experience of DCD due to the physical and cognitive demands associated with activities of daily living. It is worth noting that one participant (Ella) has co-occurring Dyslexia, however the challenges she describes above cannot easily be attributed to typical challenges associated with Dyslexia (e.g., reading and spelling). In contrast, the physical demands associated with housework, likely reflect the typical challenges associated with DCD.

**Subtheme 1.5: Challenging peer relationships in childhood.** In addition to motor skills, participants discussed challenging peer relationships in childhood as a further casualty of the symptomology associated with DCD. For example, for many the social contexts of school, and wider physical activity opportunities were problematic when participants felt that their abilities did not match the level of their peers.

*It's when I got my exam slip that I noticed that I was different to all the other people in terms of what was written on my slip and people were saying things to me like 'Oh, you get to go in the spaz room' and you're like 'you can't say that'… None of the other children wanted me on their team [in physical education]. You know, I'd be the one that they'd go 'oh, I guess we'll have her'. I was very bad at everything…* – Emily (Female, no co-occurring NDCs)

*I didn't understand the [netball] rules and none of the other girls would talk to me because they had all been on the netball team for a school year already… there was one occasion where I threw the ball into the net and it was the wrong goal and stuff like that, it was the wrong net, and I was at the wrong end. And so, then they were all furious with me because I just didn't understand that it was wrong. Yeah, I was never picked for rounders or anything just because there's no way I was going to be able to hit a ball with a bat, not something I can do* – Blake (Female, no co-occurring NDCs)

*I just never felt very graceful, and I think it kind of impacts how you feel about yourself, in a feminine way. I feel like sometimes I'm this big clumsy effort of a person… when I was a kid, I got kicked out of dance classes, which is so embarrassing. They literally told my mom, 'Oh, we don't think Katherine can come back. She's not coordinated enough'* – Katherine (Female, co-occurring Dyslexia)

Many participants noted that there was a clear differentiation between their abilities (both physically and educationally) and the abilities of their peers. This manifested in experiences of isolation and exclusion. Whilst one participant (Katherine) has a co-occurring diagnosis of Dyslexia, the challenges described above allude to stereotypical DCD presentation (e.g., clumsiness in physical activity).

**Subtheme 1.6: Commonality in difference in adult friendships.** Experiences of peer relationships within adulthood were often positioned more positively by participants. Thus, although there was evidence of a struggle to find one's place within the adult social sphere, due to not fitting in with either neurodivergent or 'neurotypical' individuals, the majority of participants spoke of commonality in difference in adult friendships. Participants experienced greater capacity to connect with those with shared experience of 'difference'.

*I've got a fair few neurodivergent friends and they're great, but I've also found it difficult because some people say 'you're too weird to be friends with the normal people'…But I don't seem the right type of weird to be friends with some of the other neurodivergent people. So, you're like, well, where do I fit in?* – Emily (Female, no co-occurring NDCs)

*Yeah, all my friends are really good. They understand what I'm good at or what I struggle with. We went walking the other week and me and my friend were holding on to each other because it was icy, and I was on my hands and knees. I'm like 'yeah, I'm not getting up there'…Sam was pushing me up the hill…and my other friend, her husband has Dyspraxia, so she understands...* – Ella (Female, co-occurring Dyslexia)

*I do have one little set of friends and it's quite funny…every single one of us is 100% neurodivergent… and actually my work colleagues, many of whom are friends, they really get it. But I also work in an environment which is very forward thinking, and we work with many people who are neurodivergent. So actually, many of us at work also identify as being neurodivergent in some way. And again, they see me trip over my bag every day and they recognise that actually this is a real thing. The number of times I spill my tea over my desk, etc.* – Izzy (Female, no co-occurring NDCs)

It is important to note that Ella has co-occurring Dyslexia, which may play a role in her experience of friendships due to shared experiences often relying on reading and processing information. However, the struggle with friendships was also evidenced in participant stories only have a DCD diagnosis. Although some participants reported struggling to 'fit in' a number of participants highlighted that they developed strong friendships within and beyond their neurodivergence with people who took the time to understand their challenges and in so doing support their needs.

**Subtheme 1.7: Commonality in difference in romantic relationships.** This commonality in difference also translated to the foundation for romantic relationships, with some participants reporting that they found love with neurodivergent partners or those who work in fields where there is a better understanding of DCD (e.g., Occupational Therapy).

*My partner has got ADHD and so, struggles with anything that requires executive function, and I can't cope in an environment where things aren't organised. Because I just need them to be. [For example] I'll come home and trip on dog toys and stuff, because Tom's been playing with the dog during the day and then he'll just leave toys wherever they fall. And I'm like 'I don't know where they are. I can't see them the same way you can because of the depth perception challenges'. The amount of times I've fully fallen over because I've tripped on like a dog bowl that's been left in a stupid place or something… We can relate to each other over both being neurodivergent in ways that I think neurotypical couples probably can't.* – Blake (Female, no co-occurring NDCs)

*Going, sort of, going into my current relationship with my husband, he's always been completely understanding and it was actually our openness about it that led eventually to him being diagnosed with ADHD because we knew that there was something that, well, quite frankly, was making him unhappy... So in some respects…we have a really great*

*relationship because we're really understanding to each other, and we can talk about it and we get it* – Amy (Female, no co-occurring NDCs)

*He [my husband] understands what it's like and he says to me 'if there's something you're struggling with then just say'. He's an Occupational Therapist so his job is to find ways around stuff. So, he says 'well, do it this way or try it this way and then you'll see if it's easier or if it doesn't make you as tired'. And he always says 'if you're too tired to do it, just don't do it and then we'll do it another day, or I'll help you when I get home from work'* – Ella (Female, co-occurring Dyslexia)

Participants revealed that support from insightful significant others allowed them to be seen, heard and understood in the context of romantic relationships. As the training for Occupational Therapy is broad and likely encompasses both DCD and Dyslexia, it is important to acknowledge that Ella's experience may be related to either of her diagnoses.

**Subtheme 1.8: Lack of equity in employment opportunities.** Beyond peer relationships in a social context, participants reported that the impact of DCD was also realised in the workplace context, on outlining the considerations central to their pursuit of employment. According to participants, many expressed a perceived lack of equity in employment opportunities which were attributed to both the seen and hidden challenges associated with DCD.

*My friend is training to be a midwife and I was like 'oh, I don't think I could do that with Dyspraxia'…. if I look at like student jobs, I definitely wouldn't be a waitress* – Katherine (Female, co-occurring Dyslexia)

*I hadn't realised quite how sensitive I am to things like noise, light. Interestingly I need lots of light, but there's a lot of people at work that like it to be dark and it really affects me. I really struggle to work in that environment and the noise is absolutely horrendous….I can't concentrate if there's lots of movement around me and lots of noise. In fact, I wrote an e-mail yesterday and…I literally transcribed half a conversation that somebody behind me was having with some-body else into this… I couldn't tune out* – Izzy (Female, no co-occurring NDCs)

*I've always done like low paid jobs down at bottom sort of thing. [I] never really achieved my potential because the sup-port wasn't there and the understanding wasn't there either* – Samuel (Male, co-occurring Autism)

Whilst two of the three quotes above relate to experiences of participants with co-occurring neurodevelopmental con-ditions, participants attributed the challenges discussed above to their DCD diagnosis, in relation to both physical skills (e.g., Midwifery) and cognition (e.g., processing information).

**Subtheme 1.9: Mental health and wellbeing across the life course.** The challenges participants faced across all life domains were reported as having burdening impact on their mental health and wellbeing, which was discussed both within the context of childhood and adulthood.

*It was awful. I had panic attacks a lot, especially like years 7–10… the panic attacks, crying all the time, feeling sick. It was just so stressful. Like I was like, 'oh, I'm getting lost'. Like it's, it's not even that big of a high school… but I would get lost and I'd just be like, 'oh, no, now I have to find my next class' and you'd be getting pushed in the corridor… I hated it.* – Ella (Female, co-occurring Dyslexia)

*Oh, I was relentlessly bullied [for] most of my life, because systems and adults often refuse to recognise that you're different, but children will. Children are horrible little things. They're like 'ah, that child is different. Let's go point it out'* – Emily (Female, no co-occurring NDCs)

*I used alcohol to cope although I have reduced this greatly in recent years. This is relevant because not only does it [DCD] impact on your mental and physical health but also on your sense of self-worth and self-identity and how you*

*view yourself within your social milieu and wider society. This can end up being a cycle of self-destruction which can be very difficult to break* – Samuel (Male, co-occurring Autism)

For many participants, stress and anxiety featured in their narratives. This sub-theme was participant-led as the interview guide did not include any specific questions pertaining to mental health and DCD. Thus, it is clear from the data that mental health and wellbeing are a key concern for this population. Whilst co-occurring diagnoses may have played a role in these experiences, it is important to note that participants reaffirmed DCD as central to the conversations outlined above.

**Subtheme 1.10: The positive role of diagnosis.**  Despite participants reporting wide-ranging secondary impacts of DCD related to physical and mental health, for many the experiences of validation through gaining a diagnosis were central to their positive experiences. Data revealed an enhanced sense of self was evident when discussing the role diagnosis played in participants' lives.

*My confidence is growing immensely since getting a diagnosis and I think it explained so much to me. There's been parts of my life prior to diagnosis where I thought I had some sort of brain injury or early onset dementia or cerebral palsy... It [DCD diagnosis] just lightens my life. I just feel so much better about myself. I can laugh at myself. I mean everything you can think of I have done. I've fallen between the gap between the train and the platform. I've set my menu on fire in a restaurant. I've slipped, putting my washing out and broken ribs. I've chipped my teeth walking down the street… But having a diagnosis is just makes all of that hilarious* – Izzy (Female, no co-occurring NDCs)

*After the diagnosis happened, I just felt like a weight had been lifted. It felt like connecting the dots and I just felt happier myself. It was like these weird things that are happening aren't me... it's the condition… It really helps me understand myself more. And I think expanding on that people understanding me more because I'm kind of more settled in who I am* – Katherine (Female, co-occurring Dyslexia)

*Being diagnosed means I understand myself a little bit better, I understand what support I need* – Blake (Female, no co-occurring NDCs)

According to participants, getting a diagnosis of DCD for participants mitigated against years of low self-worth. It allowed them to re-evaluate themselves, their identity and in so doing better understand retrospective challenges faced. This emerged as empowering for participants who appeared to engender a more positive outlook onto oneself.

**Theme 2: Lack of societal awareness, understanding and support**

Participant stories illuminated that they are not only impacted by the symptomology associated with DCD, but also arguably the profound lack of societal awareness and understanding which surrounds this underheard diagnosis. The current major theme presents evidence to substantiate further the impact of a lack of awareness, and the subsequent lack of support, across education, healthcare and the wider community. This was found to not only impact childhood but also play a significant role in lived experiences as an adult within work and wider community contexts.

**Subtheme 2.1: Non-believers in education.**  The social construction of education is one that is alleged to enable all children to thrive [35], however some participants presented contrary experiences within which they often reported negative experiences due to non-believers in this context.

*I had a few teachers who'd say, oh, I don't believe you [about DCD diagnosis and associated challenges]. I want to see, like a doctor's note, because I think you're making it up* – Christine (Female, co-occurring Autism)

*When I was at primary school, my mum tried to get me some support. The school didn't want to know. This is partly because I was hyperlexic. So, I had a reading age double my actual age* – Emily (Female, no co-occurring NDCs)

*So, I have a younger brother with moderate special needs and one of his is Dyspraxia. So, my parents were aware of it, and they saw the symptoms and they were like 'that kind of fits our eldest daughter.' I went to a really competitive girls grammar school and they kind of didn't really care not that they didn't care, but they didn't see an important thing to diagnose at that time. They were like 'we can get by without it'* – Katherine (Female, co-occurring Dyslexia)

Whilst Christine and Katherine both have co-occurring diagnoses (Autism and Dyslexia respectively) the data presented above was in direct response to whether their diagnosis of DCD impacted their experience in education (see Table 1). Despite evidential input from caregivers, and medical professionals, the burden of disbelief was evidenced in participants stories. For some, their abilities academically were positioned as a barrier to belief and ultimately support.

**Subtheme 2.2: Experiences of discrimination within the education system.** The lack of support within the education system as evidenced above, was not only presented as dismissive by participants, but also sometimes as apparent hostility which arguably reflects experiences of discrimination.

*I had this one teacher. This was when I was in year seven, so I'd have been, what, 11 years old? I was always bullied before her class, because her class was just after lunch, and she turned round to me and said 'it's rather inconvenient that you're bullied before my class because you're never here. Could you be bullied another time'* – Emily (Female, no co-occurring NDCs)

*I remember like, things like, you know, not being able to fasten like coats and get changed and being told,' oh, you're lazy, you should be able to do this by now' and I took a long time to eat physically. That's kind of challenging and like all the comments from like the staff like, 'oh, you're like this at home? Your parents must be so sick of you'… And like writing speed as well. I know that I was always very slow at writing… the teacher was like 'oh well done, this was a great assignment, but it's a shame you took too long to write it'* – Christine (Female, co-occurring Autism)

*At primary school, I didn't really have anything [support for my DCD]…. a lot of my extra stuff [support] was based on my dyslexia rather than my Dyspraxia. They didn't even speak about my Dyspraxia…I had panic attacks a lot* – Ella (Female, co-occurring Dyslexia)

Both Christine and Ella had co-occurring diagnoses of Autism and Dyslexia respectively, however it is clear from the data above that these experiences were directly related to their diagnosis of DCD, for example through mention of challenges with activities of daily living. Participants highlighted that such negative interactions were not limited to teaching staff but also support staff such as lunch time supervisors. For those with co-occurring conditions, some noted a lack of equity in support practices depending on the diagnosis.

**Subtheme 2.3: Lack of resources and support within healthcare.** According to participants, negative experiences and lack of support were also apparent within the healthcare systems which often led to them paying privately for diagnostic assessment and support.

*[After my DCD diagnosis] I was then supposed to get ongoing support. We were going to do a few sessions, but that never happened because the OT just disappeared. I don't know what happened to her.* – Blake (Female, no co-occurring NDCs)

*I tried NHS diagnosis [for DCD as an adult], the GP goes: 'we don't offer it…' I had a GP, she was good, she wouldn't accept my report because I was diagnosed privately… I feel let down by the system* – Liam *(Male, co-occurring ADHD)*

*I had to pay privately [for my DCD diagnosis]. There was no option to go through the NHS at all and I didn't actually instigate that with my GP because I know that that isn't an option. So that was already a barrier… But the assessments*

*for adults are now around I think £900, which seems completely inaccessible to most people, I think.* – Izzy (Female, no co-occurring NDCs)

Whilst some participants had co-occurring NDCs, the data presented above was in direct response a question regarding their experience of obtaining a DCD diagnosis (see Table 1). Participants highlighted challenges related to a lack of awareness amongst clinicians, timeliness of assessment and intervention as well as a lack of opportunity to engage meaningfully with healthcare as fundamental issues in their pursuit of diagnosis and support. Many participants sought private healthcare, but alluded to the associated financial implications of this.

**Subtheme 2.4: Under-supported and undervalued at work.** A lack of understanding in the social sphere, including in the workplace, peppered participants' stories. According to participants this often led to them feeling under-supported and undervalued at work, with some opting for self-employment to mitigate for this.

*I ended up leaving the Civil Service… I asked for support and the only support they could give me was a seat by the window [laughs]* – Samuel (Male, co-occurring Autism)

*Particularly the first job I had when I graduated… I can remember going and just shutting myself in a room and having a little cry because I felt like people didn't understand me.* – Izzy (Female, no co-occurring NDCs)

*I've had trouble finding and maintaining employment. I'm self-employed*- Liam (Male, co-occurring ADHD)

It is important to note that Samuel had a co-occurring diagnosis of Autism, and similarly, Liam also had diagnosis of ADHD. It is therefore important to interpret these quotes with caution as their other diagnoses may have also played a role in their experiences, however, similar experiences were described by those a sole diagnosis of DCD. It was evident within the data that employment was important to participants, however they reported that this was not always easy, due to a lack of understanding surrounding the needs of adults with DCD within the workplace. Although Samuel and Liam have co-occurring neurodevelopmental conditions which may have influenced their stories above, such experiences were commonplace amongst participants with a sole diagnosis of DCD.

**Subtheme 2.5: Misconceptions and ignorance within families.** Beyond education, healthcare and employment contexts, according to participants the lack of understanding surrounding DCD was also evident within the family context. For example, misconceptions and ignorance surrounding DCD sometimes led to participants experiencing a lack of support and interest from family members.

*My wider family are kind of like 'oh, you know, just get on with stuff'. And I'd I never really felt that I had any understanding from anybody really except my closest friends, [immediate] family, my husband.* – Amy (Female, no co-occurring NDCs)

*The wider family has definitely been more difficult…I was with my nan two weeks ago and I fell over something, and she went 'oh, you're very clumsy.' And I said 'oh, well, it's the Dyspraxia'. And she went. 'I'm so sorry to hear that. Did you just find out?' And I said 'no two years ago' and she was like, 'oh, but you must be really ill' … I think she thought it was some sort of like disease or something* – Katherine (Female, co-occurring Dyslexia)

*My mum and dad never spoke about it [DCD] at home. We never even really spoke about it at all, so it wasn't…it wasn't a thing to them… It didn't exist* – Ella (Female, co-occurring Dyslexia)

Both Katherine and Ella had a co-occurring diagnosis Dyslexia, however it is clear from the data above that these experiences were directly related to their diagnosis of DCD, for example through mention of clumsiness. Whilst participant stories reflect varying experiences within the family context, for many, negative experiences appeared to be underpinned by a lack of knowledge of DCD, which at times participants felt like this manifested as a lack of engagement with their needs.

**Subtheme 2.6: Abilities seen as not conducive to a DCD diagnosis.** Misconceptions about the challenges associated with DCD also fed into a sense of disbelief for participants when their interests, abilities and associated skillsets did not naturally align with the social construction of the DCD profile.

*Because I was a musician, and I danced, I don't think it would have occurred to anybody that I might be Dyspraxic.* – Amy (Female, no co-occurring NDCs)

*My immediate aunt and uncle were very much like 'well, Katherine went to a good school, she went to grammar school. She's at university'… So, they couldn't understand that I was struggling, or I would have these issues. And they were really quite doubtful. My parents had to be like, 'no, she's actually, genuinely struggling'* – Katherine (Female, co-occurring Dyslexia)

*It's really interesting and I think they [wider family] don't see it as a problem because they think I've achieved well in life… So, for them, it's like 'you've turned out all right… what's the issue?'* – Izzy (Female, no co-occurring NDCs)

It is important to note Katherine also had a diagnosis of Dyslexia. It is therefore important to interpret her quote with caution this may have also played a role in her experiences, however, similar experiences were described by those a sole diagnosis of DCD. There appeared to be a disconnect between participants' reality and others' expectations of what they were capable of. According to female participants, biological sex also played a role in their lived experience of denied validation due to the male-dominant diagnosis of DCD, which does not always effectively capture the female presentation.

*My mum has got a book… I think it was written in the 90s that was like 'oh, men only have Dyspraxia' and that's definitely not true. So, I think we are kind of fighting a separate fight where it's 'like women can be Dyspraxic too'* – Katherine (Female, co-occurring Dyslexia)

*Most diagnostic criteria for neurodiversity is male leading, particularly obviously under 18. So, it's boy leading. So, when you're a girl, you don't fit into those boxes. So, they go 'you're fine'* – Emily (Female, no co-occurring NDCs)

Despite advancements within DCD literature, participants' stories highlighted that the historical construction of DCD continues to impact women, whose gendered presentation can barrier support and validation.

**Subtheme 2.7: Denied validation.** Participant stories highlighted that this profound lack of awareness transcended family to wider society more broadly, resulting in denied validation. The data revealed that some were left frustrated and impotent in their challenge to be understood, whilst others felt disempowered by the apparent lack of desire to understand.

*Dyspraxia isn't still really well-known about, not really talked about… you get sick and tired of explaining it to people… I've told people, some are supportive. Some, um, some don't believe me. Some think I'm playing system, and I keep thinking, well, if I'm playing system, you know, I should be getting an Oscar [laughs]* – Samuel (Male, co-occurring Autism)

*I think the thing that frustrates me sometimes is not that people aren't aware, but people…there's just a general lack of interest, I think.* – Amy (Female, no co-occurring NDCs)

*I don't think the problem is necessarily lack of awareness in society. I think it's lack of understanding, almost importantly, lack of wanting to understand… It's when people say that they don't want to know or they say things like 'but you're just lazy', or 'everyone's just a bit clumsy'… you can't work with those sorts of people* – Emily (Female, no co-occurring NDCs)

Although Samuel had a co-occurring diagnosis, his quote directly links to a frustration around the lack of awareness surrounding DCD. Importantly, this was not a question or prompt within interview materials, and thus emerged organically

across the dataset. Participants reported that there was a lack of interest and investment in DCD at a societal level, which impacted them on an individual level, leaving some frustrated and tired from continually advocating for themselves.

**Subtheme 2.8: Inherent lack of awareness compared to other neurodevelopmental conditions.** A particular source of frustration for participants was the inherent lack of awareness across society compared to other NDCs, such as Autism, ADHD and Dyslexia, which also impacted their experience of understanding and support within healthcare, education and the wider community.

*And we're so used to talking about autism, dyslexia and ADHD. My husband has ADHD and people know straight away probably what that entails. But when you're a 45-year-old woman who has Dyspraxia, people just look at you like 'what does that mean? You can't ride a bike?'* – Amy (Female, no co-occurring NDCs)

*I don't think anybody knows what Dyspraxia is… if I'm introducing myself to a new person… I very rarely say I've got dyspraxia in the first instance because people just don't know what that means. They'll just say, 'oh, is that like dyslexia?'* – Blake (Female, no co-occurring NDCs)

*I think the trouble with dyspraxia is it's just not as well known, is it, as like, Autism and ADHD, people don't know as much about it… The awareness in the media is weak compared to ADHD* – Liam (Male, co-occurring ADHD)

All participants stories included frustration around the profound lack of awareness of DCD, especially when compared to other NDCs. As Liam had co-occurring ADHD, he therefore had an insider perspective on differentiated levels of knowledge across NDCs, in relation to their understanding and support. Many participants reported that DCD often falls in the shadows of other conditions with stronger media profiles, such as Autism, ADHD and Dyslexia.

**Subtheme 2.9: Considerable gap in community level support and advocacy.** In contrast to other well-known NDCs that have established charities and support systems within the community, participants based in the UK reflected on the considerable gap in community level support and advocacy with the closure of the Dyspraxia Foundation.

*I think that the Dyspraxia Foundation shut down at some point last year and that was a blow because I'd started to get support from them and attend events but a couple months after my diagnosis they were like 'oh, we've got bankrupt' and that's a big lost. It strikes the community because that was our big source [of support]* – Katherine (Female, co-occurring Dyslexia)

*The Dyspraxia Foundation shut down April last year… so there's no support at all, is there?* – Liam (Male, co-occurring ADHD)

*People have been left feeling like they don't know where to go for the support because actually information support is still really poor, particularly for parents with children who they think have got DCD or dyspraxia. I think they feel really isolated, and I don't think that's necessarily been filled really [since the closure of the Dyspraxia Foundation], so that is a big loss* – Izzy (Female, no co-occurring NDCs)

Whereas NDCs such as Autism, ADHD and Dyslexia are renowned for significant research and support [36]. In contrast, since the loss of the Dyspraxia Foundation (the sole charitable organisation within the UK related to DCD) [37] participants reported a sense of abandonment, with no central hub for resources and support.

## Discussion

The current research aimed to explore the reality of everyday life for individuals with DCD. The data showcased the difficult road travelled by adults with DCD both contemporaneously and retrospectively. The data transcended symptomology to wider system issues, including lack of societal awareness and effective support which impacted the lived experience

of health and wellbeing within the DCD diagnosis. Collectively, participants' accounts painted a picture of systemic disengagement – of being unheard, unsupported, and at times abandoned by the services and structures intended to help them.

**The impact of DCD**

The impact of DCD was seen across all aspects of life, as demonstrated by theme 1. For example, similarly to previous research, participants highlighted **motor skill challenges in childhood** (subtheme 1.1) played a central role in their experiences of DCD, for example some report frequently avoiding tasks that require motor skills [16]. Participants reflected upon their childhood experiences of physical activity, both structured and unstructured [5,38], which were not positive due to their challenges with motor skills. Many participants discussed the impact of **acquiring the label of 'clumsy'** (subtheme 1.2) or 'lazy', which underpinned their lived experience of DCD. Despite efforts within healthcare to move away from these outdated labels, unfortunately participants were still burdened by these, a finding which has also been echoed in other research that showcases similar denied validation in education settings [39]. Some participants reported an alarming number of injuries within childhood, in part due to trying to keep up with societal milestone expectations, i.e., learning to ride a bike, with extreme cases of families being threatened with wider service involvement. Despite a lack of research specifically on DCD and injuries in childhood, wider NDC literature alludes to a 2–3 times greater prevalence of injuries in neurodivergent children when compared to typically developing controls [40]. With a major focus on motor skills in DCD this prevalence could be higher, but more research is required to understand prevalence and subsequent implications for health and service involvement.

Physical activity was something that all participants had to engage with in childhood, due to, for example educational curricula, yet was something which participants felt created a divide between them and their peers. **Challenging peer relationships in childhood** (subtheme 1.5) were often prominent in participants stories, particularly in relation to physical activity. Many participants attributed this to their motor skill challenges which they felt influenced the perceptions and behaviours of their 'neurotypical' peers. Physical activity in adulthood was elusive within the current narrative, which could in part be explained by decreased demands due to adults no longer being beholden to educational curricula, in addition previous research has shown that withdrawal from such activities is a coping mechanism for individuals with DCD [9,16], which will result in a more sedentary lifestyle [12]. This is particularly problematic as research has suggested that low levels of physical activity in adulthood can increase the risk of obesity and cardiovascular disease [41,42]. Whilst participants attributed challenging peer relationships to DCD in the context of current interviews, literature suggests that this is not solely associated with DCD as children with other neurodivergences such as Autism and ADHD also struggle in this realm [43,44].

As evidenced in subtheme 1.9 (**Mental health and wellbeing across the life course**) participants reported challenging experiences with their peers which often resulted in participants experiencing isolation, high levels of anxiety and avoidance behaviours in relation to the school context which continued into adult life. Previous research has highlighted school-related mental health challenges as a key theme within parental narratives surrounding DCD [45].This is also in line with previous questionnaire data which highlights mental health and wellbeing challenges related to DCD; specifically related to general and movement-specific anxiety, self-efficacy, quality of life and general resilience when compared to 'typically developing' adults [46,47]. Importantly, some of the quotes in this section pertained to individuals with co-occurring NDCs, and thus some of the mental health challenges highlighted may have been as a result of DCD and/or their other diagnoses. It is well documented in the literature that children with NDCs struggle with mental health in childhood [48,49], including DCD [38,50].

Friendships in childhood have been found to be a protective factor in mental health [51]. However, parents of children with DCD identified friendships as being challenging with 'neurotypical' peers, and consequently children with DCD are often drawn to other children of difference [45]. Within the current data this was also seen in adulthood with friendships

and romantic relationships being discussed within the context of **commonality in difference in** both **adult friendships** (subtheme 1.6) and **romantic relationships** (subtheme 1.7). Although some participants highlighted a struggle to 'fit in' within the social sphere, for many, adulthood presented an opportunity to build meaningful connections in and beyond their neurodivergence. More broadly, literature suggests that neurodiverse adults tend to have more neurodiverse friends than neurotypical adults [52]. This commonality, as reflected in the current study, often leads to feelings of happiness and self-acceptance [52,53]. In contrast to the positive findings in the current study, one previous qualitative study found that adults with DCD often felt guilty due to their poor motor skills impacting on their ability to do household tasks and satisfy their partner's needs [23]. There is, however, still a lack of sufficient research focusing on romantic relationships within the realm of neurodiversity, and more specifically DCD.

Interestingly, the narrative around the impact of motor skills appeared to shift when talking about adulthood, moving beyond physical activity through play [54], to more functional skills and **developmental milestones which enable independence** (subtheme 1.3) in adulthood, such as driving [11]. Previous research has reported that individuals with DCD find learning to drive challenging [12]. Many participants in the current study reported that passing their test took multiple attempts and years of lessons. Even after passing, some participants suggested that they were still not confident with certain aspects of driving such as parking and navigating due to poor spatial awareness, which are challenges evidenced in previous literature [10,55,56].

Despite there being no specific interview questions or prompts on cognition, this was often central to participants' stories. For example, participants illuminated the burden of the high **cognitive load** (subtheme 1.4) required to drive both competently and confidently [36]. Participants also attributed cognitive challenges to their lived experience of DCD more broadly, for example in relation to the planning and processing involved in daily functionality, particularly when participants had busy family lives. According to participants, such challenges resulted in them making errors such as forgetting appointments, and arriving late, whilst also impacting upon their ability to keep up with the pace of modern life. Previous research suggests that these underheard challenges associated with DCD, ones that don't appear on diagnostic criteria can have a significant impact the day to day lives of individuals with DCD [9,10,13,46]. The current data illuminates how exhausting DCD is for participants due to how hard they have to work to sustain their role(s) in life and society. It is important to note that there was co-occurrence of other NDCs amongst participants, and therefore on interpreting the impact of cognitive load, for some their other diagnoses may have also played a role. For example, previous literature shows that individuals with Autism [57–59], ADHD [59,60] and Dyslexia [61,62] (as per some of the participants in this study) also struggle with cognition. Future research would benefit from disentangling cognitive challenges amongst NDCs.

According to Wenger (1998), who we are is underpinned by experiences of purpose and meaning through participation within the communities of which we belong [63,64]. A recognised definer of identity in adulthood is one's employment [65]. Participants expressed a **lack of equity in employment opportunities** (subtheme 1.8), which was defined by the considerations made when considering career options. Research suggests that there is concern amongst parents of young adults with DCD regarding their child's capacity to function within the workplace [9]. Conversely, meaningful employment has been found to have a positive impact on mental health in DCD populations [20]. Participants revealed that both the seen and hidden challenges related to DCD influenced their thinking surrounding employment [66]. Some participants ruled out, for example, manual jobs, or skilled roles which were reliant on motor skills, such as midwifery, whilst others avoided roles which heavily relied on cognitive abilities such administration, or environments such as open plan offices that participants felt were not conducive to coping with their lived experience of DCD. This aligns with previous research showing that cognition may have a particular impact on employment for young adults with DCD [12]. This has also been found in adults with other NDCs e.g., Autism [67–69] for which some of the participants in the study have co-occurring diagnoses, thus it is important to note that some of their experiences may also be attributed to their diagnosis of another NDC. However, participants with DCD alone also reported such challenges.

Despite the many challenges participants faced in relation to their DCD, the **positive role of diagnosis** (subtheme 1.10) often appeared transformational. Experiences of validation were illuminated throughout participants stories when discussing the role diagnosis played in their lives and sense of self. The power of diagnosis has previously been evidenced in neurodevelopmental condition literature more broadly [70–72]. However, the current research provides evidence that a diagnosis plays a role in quality of life and sense of self in DCD. This mirrors previous research in wider neurodevelopmental conditions which highlights the relationship between diagnosis and self-acceptance through a validating identity [70–72].

### Lack of societal awareness, understanding and support

A second major theme that emerged as a result of participant discussions was a lack of awareness and support across all sectors of society including education, healthcare, families and the wider community. Participants highlighted that this did not only impact childhood but also played a significant role in their experiences as an adult. Within the context of childhood many participants reported negative experiences due to **non-believers in education** (subtheme 2.1). For example, participants claimed that some teachers demanded to see doctor's notes for proof, whilst other teachers were blinded by what participants could do which denied any acknowledgement of what they could not do. Unfortunately, non-believers are commonplace for individuals with invisible disabilities [73]. In the school context, teacher support has been found to play a pivotal role in student success [74] meaning some children with DCD may not only barriered by symptomology but also by a lack of supportive scaffolding. This lack of support reflected participants' **experiences of discrimination within the education system** (subtheme 2.2). For example, being told that their parents must be sick of them (for being slow with dressing and eating) and expressing frustration for regularly missing a class due to incessant and relentless bullying. Although previous literature with young adults with suspected DCD alludes to experiences of bullying and teasing from peers [16], the current data suggests that there is potential for this to extend to those in a position of power, e.g., from teachers. It is however important to note that many participants attended school 20–30 years ago, and although general acceptance of neurodivergence has increased [75], the current data reveals the risk of this happening is real due to the persistent lack of awareness surrounding DCD.

These negative experiences were also transparent within participants' reflections on the **lack of resources and support within the healthcare** systems (subtheme 2.3) which often led to participants paying privately for a diagnosis and support. Although for some participants private healthcare was a viable option, for many, particularly those from lower socioeconomic backgrounds, this would not be possible [76,77], which has the potential to exacerbate health inequalities [76]. For some participants, their needs were missed as a child and thus had to rely on private diagnosis in adulthood. As there is a lack of a standardised pathway for assessment beyond childhood [78], for these participants the only option was to pay to get the diagnosis and support they needed. Participants also reported a lack of person-centred care and intervention, leaving them feeling let down by public healthcare. Research has illuminated many barriers within the current pathways for DCD assessment and support including a lack of awareness, long waiting lists and disconnected systems and services [79–81], and thus their discontent is understandable.

Beyond education and healthcare contexts, participants highlighted that the lack of understanding and support surrounding DCD was also evident within **misconceptions and ignorance within families** (subtheme 2.5). For example, similarly to within the education system where some participants were arguably abandoned due to the blinding effect of what they were able to do, misconceptions and ignorance surrounding DCD sometimes led to participants experiencing a lack of support and interest from family members. This is consistent with the literature which reflects the reality of an underheard diagnosis being problematic within the family context when there is a profound lack of interest amongst family members which realises itself in a lack of support and validation [82]. For some participants this lack of knowledge, awareness and interest in DCD led to them not receiving a diagnosis of DCD until much later in life. Early intervention is

known to be effective in DCD [83], but is often not possible due to many healthcare services internationally relying on a diagnostic-led approach [84], which means support cannot be accessed prior to a diagnosis being given [85].

Participants reported that a lack of inclusive practices often led to feeling **under-supported and undervalued at work** (subtheme 2.4). For some participants, this resulted in them pursuing self-employment to ensure that their needs were met. Although there is a paucity of research specifically exploring the role of self-employment in DCD, the literature suggests that self-employment is beneficial for young adults with neurodevelopmental conditions as this enables social and economic participation [86,87]. Given the UK Government's current strategy to 'Get Britain Working' [21], there is scope and impetus to improve support for people with DCD in the workplace. A collaborative approach that encompasses increased awareness and understanding could be used to develop frameworks for workplaces, as has been implemented previously for mental health conditions, such as creating more mentally healthy workplaces [88]. This, however, needs to be undertaken sensitively, as although there is greater acceptance and encouragement of diversity in the workplace [89], this has not always necessarily been extended to neurodivergence [90].

**Abilities not seen as conducive to a DCD diagnosis** (subtheme 2.6) were also evident in participant stories. Whilst DSM-5 criteria [4] state that the acquisition and execution of motor skills is below age expected standards, for many this meant that their needs were sometimes not recognised. For example, Amy learned to play the piano from a young age and uses this skill within her work now, however her proficiency in one skill overshadowed her challenges with other motor skills in daily living. Literature shows that individuals with DCD can develop proficiency in specific practiced motor skills [6], although participants noted that this was often not acknowledged amongst family, friends and some clinicians.

Participants were particularly frustrated by the **inherent lack of awareness compared to other neurodevelopmental conditions** (subtheme 2.8), with specific references made to Autism, ADHD and Dyslexia. Despite Autism being approximately five times less prevalent than DCD [29,91], it became clear that participants felt their diagnosis was in the shadows in comparison. Literature validates these feelings, with healthcare professionals and education professionals having poorer knowledge of DCD compared to Autism [92]. For some, the lack of awareness was also exacerbated by the fact that the wider community seemed to have little interest in understanding their challenges, resulting in a lack of support options available. Within the UK, participants claimed that there was a **considerable gap in community level support and advocacy** (subtheme 2.9) as a result of the closure of the Dyspraxia Foundation [37] which removed the main source of information for individuals with DCD and their families. A search of the Charity Commission for England and Wales revealed that there are 150 established charities that have Autism in their name [93]. This disparity in support, as perceived by participants, reinforces a sense that DCD remains isolated and overlooked within the broader neurodevelopmental landscape. This lack of awareness and support compared to other neurodevelopmental conditions sometimes left participants feeling frustrated and like they had been **denied validation** (subtheme 2.7), which also manifested for some participants as an evident lack of interest in their lived experience of DCD.

Taken together, participants' experiences point to a need for evidence-based training across sectors (healthcare, education, and the wider community) to enable greater awareness and support for individuals with DCD. Importantly, this training should highlight not only challenges within childhood, but also how these challenges manifest within adulthood. It is, however, important that such training is co-produced with lived experience to ensure the information presented effectively captures the totality of DCD. In addition, this study adds to the literature showing a lack of capacity within healthcare to assess and support neurodevelopmental conditions in a timely manner [94,95], leading to many families opting to pursue private healthcare.

## Limitations and future directions

It is important to note that findings reflect the experiences of a small qualitative sample ($n = 10$) and therefore may not be representative of all adults with DCD. Particularly given funding was not available to supply interpreters for this research, and thus only participants who were fluent in English had an opportunity to take part. This could result in the data

presented being reflective of a Westernised view of DCD, which has potential to skew findings. Future research should endeavour to include lived experience from a wider range of countries, to better understand the global picture. In addition, half of the sample had co-occurring Autism, ADHD or Dyslexia. Although this reflects the reality of DCD diagnoses, in which there are often one or more co-occurring NDC diagnoses, [14] this makes it challenging to determine whether experiences were primarily attributable to DCD, co-occurring conditions, or their interaction. However, participants attributed these experiences to their DCD diagnosis, as opposed to any co-occurrence, so it is important to acknowledge these claims within the realm of lived experience.

There was also an unequal gender split for participants, with only two males taking part. This is not reflective of the DCD population more broadly [96,97], and future research should endeavour to capture greater equity in gendered experiences. There was also an unequal split between UK-based (nine participants) and international (one participant who grew up in the UK) interviews, despite endeavours to recruit across English speaking countries to explore cultural differences. Due to there only being one international participant, cross-cultural comparison and analysis was not possible. Future research would benefit from gathering a wider international perspective to explore whether experiences differ across countries.

In addition, participant age was not obtained through the interviews. This limits our ability to situate participants' retrospective accounts within specific historical or generational contexts. Without age data, we cannot determine whether variations in reported experiences reflect generational differences, for example societal attitudes, or individual differences within the same timeframe. Additionally, the time between participants' experiences and the interview is unknown, which may affect both memory accuracy and how participants frame their narratives. Future research would benefit from collecting age data to enable more nuanced interpretation of how experiences may have changed across cohorts and time periods. Formal evidence of a DCD diagnosis was not obtained from participants as for many, DCD is diagnosed in childhood. Despite this lack of formal evidence, all participants were asked to discuss their experiences of diagnosis (see Table 1). For those diagnosed later in life, comprehensive accounts of their diagnostic journey were shared, however for some, who were diagnosed in early childhood, such memories were underpinned by parental narrative. One way future research could mitigate for the potential lack of access to official diagnosis records is by using the Adult DCD Checklist (ADC; [98]). Despite the lack of national and international consensus around how to assess DCD in adults, the ADC offers an evidence-based solution [98], which could mitigate for the lack of standardised pathways for adult diagnosis (which has the potential to limit the pool of eligible participants). Finally, future research would benefit from incorporating the voice of the child to capture first-hand accounts of the challenges faced within childhood, as opposed to retrospectively. It would also be beneficial to take a longitudinal approach to capture how the DCD landscape shifts with age along the lifespan.

## Conclusion

The current study identified two major themes and 19 subthemes that collectively illuminate the lived experience of adults with DCD across both childhood and adulthood. Data revealed that the impact of DCD extended well beyond the motor skill challenges central to diagnostic criteria, encompassing the cognitive load of everyday activities, a perceived lack of equity in employment opportunities, and wide-ranging impacts on mental health and wellbeing across the life course. Alongside this, a second major theme highlighted a profound and pervasive lack of societal awareness, which resulted in a lack of support, evident across education, healthcare, family contexts, and the wider community. Together, these findings indicate that individuals with DCD often feel alone, isolated and misunderstood in a world which presents challenges for them across all facets of life. These findings suggest an urgent need for the profile of DCD to be raised, however, it is imperative that insider voices play a key role in these discussions, to ensure their experiences are accurately reflected. In addition, more needs to be done to ensure a lifespan approach to DCD, including greater scaffolding and support within education, healthcare and the workplace. If adults with DCD are adequately supported within wider society, there is reason to believe there will be greater opportunities for them to thrive alongside their 'neurotypical' peers.

## Supporting information

**S1 File. COREQ Checklist.**

(PDF)

## Author contributions

**Conceptualization:** Rebecca Murray, Cara E. Staniforth.

**Data curation:** Rebecca Murray, Cara E. Staniforth, Lucy H. Eddy.

**Formal analysis:** Rebecca Murray, Cara E. Staniforth, Lucy H. Eddy.

**Funding acquisition:** Lucy H. Eddy.

**Investigation:** Rebecca Murray, Cara E. Staniforth, Lucy H. Eddy.

**Methodology:** Rebecca Murray.

**Project administration:** Rebecca Murray, Cara E. Staniforth, Lucy H. Eddy.

**Resources:** Rebecca Murray, Cara E. Staniforth, Lucy H. Eddy.

**Supervision:** Rebecca Murray, Lucy H. Eddy.

**Validation:** Rebecca Murray, Lucy H. Eddy.

**Writing – original draft:** Rebecca Murray.

**Writing – review & editing:** Rebecca Murray, Cara E. Staniforth, Lucy H. Eddy.

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
