## [Decision Letter · Decision Letter 0]

24 Sep 2025

PONE-D-25-33989‘The world is just so fast, and I’m not fast… it’s just really, really difficult to keep up’: A qualitative exploration of the lived experience of adults with Developmental Coordination DisorderPLOS ONE

Dear Dr. Murray,

Thank you for submitting your manuscript to PLOS ONE. I do want to personally apologize for the delay in returning the results of review. I had a very difficult time securing two reviewers. After careful consideration, we feel that it has merit but does not fully meet PLOS ONE’s publication criteria as it currently stands. Therefore, we invite you to submit a revised version of the manuscript that addresses the points raised during the review process. The topic of your paper is potentially valuable and important. However, both reviewers have made very helpful comments that should strengthen the manuscript. They also offered other important suggestions and I urge you to attend to all of them. I look forward to a revised version of the paper.

We look forward to receiving your revised manuscript.

Kind regards,

Yu-Wei Ryan Chen, PhD

Academic Editor

PLOS ONE

Journal Requirements:

“The work of the senior author (L.H. Eddy) and co-author (C.E. Staniforth) was supported by a grant from the Waterloo Foundation (ref: 27665413). “

3. In the online submission form, you indicated that “Due to the sensitive nature of lived experience interviews, the research team believe that it would be inappropriate to make the data available. However, if anyone requests the data, we will provide heavily redacted transcripts - fully anonymised to ensure participant anonymity.”

Reviewers' comments:

Reviewer's Responses to Questions

**Comments to the Author**

1. Is the manuscript technically sound, and do the data support the conclusions?

Reviewer #1: Yes

Reviewer #2: Partly

2. Has the statistical analysis been performed appropriately and rigorously? 

Reviewer #1: Yes

Reviewer #2: Yes

3. Have the authors made all data underlying the findings in their manuscript fully available?

Reviewer #1: Yes

Reviewer #2: No

4. Is the manuscript presented in an intelligible fashion and written in standard English?

Reviewer #1: Yes

Reviewer #2: No

5. Review Comments to the Author

Reviewer #1: This paper qualitatively investigates the lived experience of adults with DCD, an important and underexplored topic. The study addresses a gap in the literature, and several aspects—such as the reflexive approach and consideration for participant comfort—are commendable. However, there are areas that require further development, particularly in the Results section, before the manuscript is ready for publication. Below, I outline my general comments and specific suggestions for improvement.

General comments

Throughout the paper, I recommend using the term “condition” instead of “disorder” (except when referring to DCD by its full name, as this is the official diagnostic term). Similarly, “co-occurrence” is preferable to “co-morbidity.”

The manuscript notes that participants were recruited cross-culturally, which is a strength, but this perspective is not fully developed in the aims, research questions, results, or discussion. Please expand on why cultural differences might matter for DCD experiences and reflect these differences in the analysis.

The phrase “formally known as Dyspraxia” could be misleading. Dyspraxia is still widely used in educational, healthcare, and employment contexts, whereas DCD is primarily used in research and has formal diagnostic criteria (unlike Dyspraxia). Please clarify the diagnostic terminology to avoid confusion.

Abstract

Include the mean age and SD of participants

Intro

As DCD is primarily a movement condition, please introduce this before discussing sensory integration difficulties, aligning the description with the four DSM-5 diagnostic criteria.

Where possible, cite original research evidence for the persistence of DCD into adulthood, rather than secondary recommendations seen in the EADC document.

At times, the literature is presented as a list of findings without deeper synthesis. Please use the literature to build an argument that logically narrows toward the research gap. Here, findings are introduced quickly and sequentially without enough critical commentary to connect them.

The rationale for taking a lifespan, cross-cultural qualitative approach is mentioned but not sufficiently unpacked. There’s no elaboration on why cross-cultural recruitment might reveal different experiences, or how qualitative interviews address the identified research gap better than other methods.

Add the research questions in line with APA recommendations for qualitative research.

Clarify abstract concepts (e.g., “moving through space,” “illusive resolution”) with more concrete language.

Acknowledge that DCD can be diagnosed at any age.

Where statements imply direct causation (e.g., challenges directly linked to poorer well-being), please expand and clarify.

The sentence “Although many of these studies have focused on university populations” should be supported with a reference.

Methods

As there were only 7 questions used in this study, I recommend including key questions (opr all of them) in the main script to make the study’s focus clear to readers.

Provide more detail about the diagnostic evidence you used for the participants, what age were they diagnosed, who by and in which country.

Include justification for sample size with reference to previous research.

Age, gender, and country breakdown are provided, but not linked to the research aims.

It is a strength that interview questions were sent to some participants in advance and that member checking was used—this demonstrates a participant-centred, reflexive approach.

Results

This section contains valuable, vivid participant accounts, but requires reorganisation for clarity and alignment with APA style. While this section provides a rich array of direct participant quotations, it currently reads as an extended list of personal anecdotes rather than a thematically organised, reader-friendly presentation of findings. This approach makes it difficult to see how each quotation ties directly to the identified themes or subthemes, and it does not fully align with APA style guidelines for reporting qualitative results.

Introduce all themes and subthemes at the start of the Results section, ideally with a visual representation showing their interrelationships.

Begin each theme with a short analytic summary (1–3 sentences), followed by one or two representative quotes.

Embed quotations within the narrative and provide interpretation immediately after each quote to link it back to the research questions or relevant literature.

Use concise, well-chosen excerpts rather than multiple long quotes on the same issue—this will reduce repetition and increase impact.

Add demographic details after each quote (e.g., “Ella, female, 27, New Zealand”) to help readers interpret variation.

Check for even distribution of quotes across participants; if some provided richer data, summarise rather than repeat multiple long extracts from the same individual.

Consider whether the word “treacherous” in the introductory sentence is the most appropriate descriptor.

Discussion

Strengthen the link between discussion points and clearly defined research questions and themes/subthemes.

You have mentioned the cross-cultural element but there is no reference to differences in the results or discussion – please elaborate for the reader

Where practical implications are discussed, provide more specific recommendations for interventions or policy.

Include a brief reflexive statement on how researcher positionality may have influenced interpretation.

Whilst you mention this in the limitations, some more information on how you unpacked the influence of the co-occurrences of DCD with ASD, ADHD would be helpful so that the reader can be confident that the findings are the result of DCD and not co-occurrences.

In the first paragraph, “explore” may be preferable to “reveal” to avoid implying preconceived bias.

Clarify the meaning of “transcending symptomology.”

Limitations

Given that you used a diagnosis of DCD as your inclusion criteria, this should be discussed as a limitation as there is so little known about DCD and no nationally or internationally enforced guidelines for diagnosis. Whilst using the ADC is not perfect, it provides a standardised tool to ensure the ppts all meet the criteria for DCD in a research setting, discuss the implications of this criterion.

Reviewer #2: Summary

Thank you for the opportunity to review this important manuscript. This manuscript makes an important contribution by documenting the lived experiences of adults with DCD. It highlights the significant role that lack of awareness and support from families, schools, workplaces, and broader society plays in shaping the quality of life, mood, and self-esteem of individuals with DCD. It also highlights important implications clincially but also for research and policy.

Strengths

- Provides valuable insight into the lived experience of adults with DCD, drawing on participants’ reflections across both childhood and adulthood.

- Brings needed attention to an under-researched area, with clear implications.

That being said, the manuscript would benefit from revisions in some areas:

- Although the authors acknowledge that co-occurring neurodevelopmental (ND) conditions were present in approximately 50% of the sample, the analysis does not adequately disentangle the impact of DCD alone versus the cumulative impact of DCD plus other ND conditions.

- As a result, conclusions attributing outcomes such as cognitive load, concentration, and tiredness exclusively to DCD per se appear overstated. These domains may also be shaped independently or cumulatively by ADHD, autism, or mood symptoms. Tiredness/exhaustion can be attributed to mood symptoms as well which may be a consequence of experiencing DCD (and its impacts due to lack of awareness), or may be independent of DCD, or may be a result of the co-occurrence of DCD and other ND.

- The limitations section briefly notes this issue, but a more substantial engagement with the implications for interpreting results is warranted.

Major Issues

- Clarity of “lack of awareness”: In the abstract and early sections, it is initially unclear whether the “lack of awareness” refers to participants’ own awareness of DCD or the lack of awareness in their surrounding environments (families, schools, workplaces, and society). Clearer wording in the abstract and main body would prevent this ambiguity.

- Claim of cultural differences: With only 10 participants from the UK and NZ, the study cannot claim meaningfully that cultural contexts were compared. This claim was not discussed in the discussion section either. Please adjust wording or clarify limitations.

- Attribution of outcomes to DCD alone: Results and discussion sections sometimes present concentration, cognitive load, and tiredness as direct consequences of DCD. Given that 50% of participants reported co-occurring ND conditions (ADHD, autism, dyslexia), these effects may reflect cumulative neurodivergence. Please temper causal claims and/or clarify that these are participant attributions. Greater consideration of the potential role of co-occurring ND conditions in these areas would strengthen the validity of the conclusions.

- Handling of comorbidity: While noted in the limitations, comorbidity should be integrated more explicitly into interpretation of results. It may be beneficial to flag the quotes by comorbidity status after the name or providing sensitivity commentary to help readers judge which findings are more specific to DCD.

- Methods/Sample size rationale: Ten interviews are appropriate. However, a brief rationale may be required (e.g., information power, iterative analysis) as well as a statement on how saturation was judged to justify why 10 participants were included (and not more or less).

Minor Issues

- Several typos, grammatical errors, and issues with punctuation were noted. A careful proof-read is recommended.

- Abbreviations were noted that were not written in full in the first instance (Eg EACD) and would be helpful for the reader if presented appropriately in the manuscript.

- Use of causal language should be reconsidered as DCD may not be the only reason for the cognitive load/tiredness.

Recommendation

- This is important work that deserves publication. However, the concerns outlined above, particularly regarding the interpretation of findings in light of co-occurring conditions and clarity of language, warrant major revisions.

6. PLOS authors have the option to publish the peer review history of their article (what does this mean?). If published, this will include your full peer review and any attached files.

Reviewer #1: **Yes:** Judith Gentle

Reviewer #2: No

---

## [Author Response · Author response to Decision Letter 1]

18 Nov 2025

Reviewer 1

This paper qualitatively investigates the lived experience of adults with DCD, an important and underexplored topic. The study addresses a gap in the literature, and several aspects—such as the reflexive approach and consideration for participant comfort—are commendable. However, there are areas that require further development, particularly in the Results section, before the manuscript is ready for publication. Below, I outline my general comments and specific suggestions for improvement.

We thank the reviewer for their kind words. We really appreciate the time they have taken to review the manuscript. We have worked through all of their suggestions and believe the manuscript is much stronger for their input.

General comments

Throughout the paper, I recommend using the term “condition” instead of “disorder” (except when referring to DCD by its full name, as this is the official diagnostic term). Similarly, “co-occurrence” is preferable to “co-morbidity.”

We agree this terminology is preferable and thus have changed this throughout the manuscript.

The manuscript notes that participants were recruited cross-culturally, which is a strength, but this perspective is not fully developed in the aims, research questions, results, or discussion. Please expand on why cultural differences might matter for DCD experiences and reflect these differences in the analysis.

The rationale for taking a lifespan, cross-cultural qualitative approach is mentioned but not sufficiently unpacked. There’s no elaboration on why cross-cultural recruitment might reveal different experiences, or how qualitative interviews address the identified research gap better than other methods.

Although we advertised the study to all English speaking countries, we were only able to recruit one individual from overseas (New Zealand). This participant grew up in the UK - we have not mentioned this as they wanted this to be redacted to ensure anonymity. They were therefore reflecting on experiences within the UK (childhood) and New Zealand (adulthood). One international participant was not enough to explore cross cultural differences, so we have removed all mention of this from the manuscript. We have added some blurb about this to the limitation section.

The phrase “formally known as Dyspraxia” could be misleading. Dyspraxia is still widely used in educational, healthcare, and employment contexts, whereas DCD is primarily used in research and has formal diagnostic criteria (unlike Dyspraxia). Please clarify the diagnostic terminology to avoid confusion.

We have amended the transcript to reflect that DCD is the diagnostic term, however it is also known as and referred to as Dyspraxia.

Abstract

Include the mean age and SD of participants

This information was not collected from participants. We appreciate this information may have been helpful in determining generational changes in experience – so we have included this as a limitation in the discussion.

Intro

As DCD is primarily a movement condition, please introduce this before discussing sensory integration difficulties, aligning the description with the four DSM-5 diagnostic criteria.

We have removed the sensorimotor integration aspect to remove confusion. We have instead listed the diagnostic criteria from the DSM-V.

Where possible, cite original research evidence for the persistence of DCD into adulthood, rather than secondary recommendations seen in the EADC document.

We have included a sentence about this being reflected in academic literature too. Further discussion of this literature is also later on in the introduction.

At times, the literature is presented as a list of findings without deeper synthesis. Please use the literature to build an argument that logically narrows toward the research gap. Here, findings are introduced quickly and sequentially without enough critical commentary to connect them.

We have endeavoured to restructure the introduction to allow for greater synthesis and direction towards the research gap.

Add the research questions in line with APA recommendations for qualitative research.

We have aligned the paper to the COREQ checklist for qualitative research as per PLOS recommendations. The checklist for this can now be found in supplementary material 1.

Clarify abstract concepts (e.g., “moving through space,” “illusive resolution”) with more concrete language.

We have changed moving through space to “being able to successfully navigate their environment”, and illusive resolution to “which were not easily overcome”. We hope this clarifies the intended meaning within the manuscript.

Acknowledge that DCD can be diagnosed at any age.

We agree this was unclear in the manuscript. We have added a caveat to the diagnosis section which states “DCD is often, although not exclusively diagnosed in childhood”.

Where statements imply direct causation (e.g., challenges directly linked to poorer well-being), please expand and clarify.

We have amended the language in both the results and discussion to mitigate for implied causality. Rather the changes allude to relationships shown in the literature.

The sentence “Although many of these studies have focused on university populations” should be supported with a reference.

We have now included references to justify the inclusion of this sentence.

Methods

As there were only 7 questions used in this study, I recommend including key questions (or all of them) in the main script to make the study’s focus clear to readers.

We have now included these as a table (with prompts) in the main manuscript (Table 1). We agree this will facilitate easier deciphering of findings.

Provide more detail about the diagnostic evidence you used for the participants, what age were they diagnosed, who by and in which country.

We did not ask for evidence of a formal diagnosis to be provided for this study, as for many, DCD is diagnosed in childhood. As we were interviewing adults, there were concerns that participants may not be able to access formal evidence of a diagnosis. We have however acknowledged this as a limitation in the discussion section.

Include justification for sample size with reference to previous research.

We have justified the sample size in the participants section of the methods, through both previous literature, and our experience of data saturation.

Age, gender, and country breakdown are provided, but not linked to the research aims.

Participant demographics were provided to give insight into the participants that were recruited, rather than these being linked directly to research aims (other than age – for which we were looking specifically at the adult population). As such, we have referenced these demographics in the discussion but not specifically analysed the data on this basis. Particularly as this would not be possible for many of these demographics (e.g. sex where there were only two males, and country, where only one was international, and this participant grew up in the UK).

It is a strength that interview questions were sent to some participants in advance and that member checking was used—this demonstrates a participant-centred, reflexive approach.

We thank the reviewer for their kind words, we wanted to make sure all participants felt their stories were a true reflection of their experiences, and we were grateful for their input through member checking.

Results

This section contains valuable, vivid participant accounts, but requires reorganisation for clarity and alignment with APA style. While this section provides a rich array of direct participant quotations, it currently reads as an extended list of personal anecdotes rather than a thematically organised, reader-friendly presentation of findings. This approach makes it difficult to see how each quotation ties directly to the identified themes or subthemes, and it does not fully align with APA style guidelines for reporting qualitative results. Introduce all themes and subthemes at the start of the Results section, ideally with a visual representation showing their interrelationships.

We have re-organised the results section to clarify major themes and subthemes via headings. We have also included a table of themes (Table 2) at the start of the results section to guide the reader.

Begin each theme with a short analytic summary (1–3 sentences), followed by one or two representative quotes. Embed quotations within the narrative and provide interpretation immediately after each quote to link it back to the research questions or relevant literature.

We have re-structured the results section to ensure there is a short analytic summary at the start of each theme, and subtheme, as well as some interpretation of the quotes after their presentation. Further interpretation, including links to literature is presented within the discussion section, so we did not include this in the Results section to avoid duplication. In the results section, we also ensured that where necessary co-occurring neurodevelopmental conditions were acknowledged and discussed in relation to the subthemes.

Use concise, well-chosen excerpts rather than multiple long quotes on the same issue—this will reduce repetition and increase impact.

Literature applauds the use of longer quotes which effectively illustrate lived experience of the phenomenon under investigation (Lingard 2019). However, we have reviewed all quotes and reduced (in length and quantity) where necessary. As the participant voice was central to the current research, we placed significant value in the raw complexities of participants lived experiences as presented within the results. Thus, deleting parts of quotes has the potential to undermine what mattered to participants.

Add demographic details after each quote (e.g., “Ella, female, 27, New Zealand”) to help readers interpret variation.

We have added the demographic details we have and can share to the names after each quote. As only one participant was based internationally, and was concerned about her family finding out about her interview. We therefore worked with her to understand what she would be comfortable with, and we agreed to not disclose her location. In addition, age was not collected.

Check for even distribution of quotes across participants; if some provided richer data, summarise rather than repeat multiple long extracts from the same individual.

We have checked for distribution across participants and removed some data from those who appeared more frequently. We have removed any repetition from the same individual in each subtheme. There are two participants (Danielle and Christine) who have much fewer quotes presented, however, the quality of interviews dictated to some extent the inclusion of quotes. However, we made sure that all participants were included in the final manuscript. Unfortunately, the two aforementioned interviews did not talk as freely as others, which limited their contribution to the development of themes.

Consider whether the word “treacherous” in the introductory sentence is the most appropriate descriptor.

We have changed treacherous to “difficult” for ease of clarity.

Discussion

Strengthen the link between discussion points and clearly defined research questions and themes/subthemes.

The discussion has been restructured to clearly map each subtheme to relevant literature. We have ensured all subthemes are present and discussed in relation to other studies.

You have mentioned the cross-cultural element but there is no reference to differences in the results or discussion – please elaborate for the reader

We apologise for the inclusion of this in the abstract. Unfortunately, there was only one international participant, so cross-cultural analysis and comparison was not possible. We have mentioned this in the discussion (limitations) section.

Where practical implications are discussed, provide more specific recommendations for interventions or policy.

We have made specific recommendations in relation to co-producing training, and re-organising pathways for assessment and support in DCD.

Include a brief reflexive statement on how researcher positionality may have influenced interpretation.

We have included a section on the research team and reflexivity at the end of the methods section.

Whilst you mention this in the limitations, some more information on how you unpacked the influence of the co-occurrences of DCD with ASD, ADHD would be helpful so that the reader can be confident that the findings are the result of DCD and not co-occurrences.

In the results section, we ensured that where necessary co-occurring neurodevelopmental conditions were acknowledged and discussed in relation to the subthemes. We have also checked the discussion section to ensure that all results discussed are relevant to DCD.

In the first paragraph, “explore” may be preferable to “reveal” to avoid implying preconceived bias.

We thank the reviewer for this suggestion. It has now been implemented.

Clarify the meaning of “transcending symptomology.”

We have changed the phrasing of this within text to “extended beyond symptomology”.

Limitations

Given that you used a diagnosis of DCD as your inclusion criteria, this should be discussed as a limitation as there is so little known about DCD and no nationally or internationally enforced guidelines for diagnosis. Whilst using the ADC is not perfect, it provides a standardised tool to ensure the ppts all meet the criteria for DCD in a research setting, discuss the implications of this criterion.

We agree the ADC could be useful in this context – we have included this within the limitations and future directions section.

Reviewer 2

Thank you for the opportunity to review this important manuscript. This manuscript makes an important contribution by documenting the lived experiences of adults with DCD. It highlights the significant role that lack of awareness and support from families, schools, workplaces, and broader society plays in shaping the quality of life, mood, and self-esteem of individuals with DCD. It also highlights important implications clinically but also for research and policy.

Strengths

- Provides valuable insight into the lived experience of adults with DCD, drawing on participants’ reflections across both childhood and adulthood.

- Brings needed attention to an under-researched area, with clear implications.

We thank the reviewer for their kind words.

That being said, the manuscript would benefit from revisions in some areas:

- Although the authors acknowledge that co-occurring neurodevelopmental (ND) conditions were present in approximately 50% of the sample, the analysis does not adequately disentangle the impact of DCD alone versus the cumulative impact of DCD plus other ND conditions.

- As a result, conclusions attributing outcomes such as cognitive load, concentration, and tiredness exclusively to DCD per se appear overstated. These domains may also be shaped independently or cumulatively by ADHD, autism, or mood symptoms. Tiredness/exhaustion can be attributed to mood symptoms as well which may be a consequence of experiencing DCD (and its impacts due to lack of awareness), or may be independent of DCD, or may be a result of the co-occurrence of DCD and other ND.

-The limitations section briefly notes this issue, but a more substantial engagement with the implications for interpreting results is warranted.

- Attribution of outcomes to DCD alone: Results and discussion sections sometimes present concentration, cognitive load, and tiredness as direct consequences of DCD. Given that 50% of participants reported co-occurring ND conditions (ADHD, autism, dyslexia), these effects may reflect cumulative neurodivergence. Please temper causal claims and/or clarify that these are participant attributions. Greater consideration of the potential role of co-occurring ND conditions in these areas would strengthen the validity of the conclusions.

- Handling of comorbidity: While noted in the limitations, comorbidity should be integrated more explicitly into interpretation of results. It may be beneficial to flag the quotes by comorbidity status after the name or providing sensitivity commentary to help readers judge which f

---

## [Decision Letter · Decision Letter 1]

25 Dec 2025

PONE-D-25-33989R1‘The world is just so fast, and I’m not fast… it’s just really, really difficult to keep up’: A qualitative exploration of the lived experience of adults with Developmental Coordination DisorderPLOS One

Dear Dr. Murray,

Thank you for submitting your manuscript to PLOS ONE. After careful consideration, we feel that it has merit but does not fully meet PLOS ONE’s publication criteria as it currently stands. Therefore, we invite you to submit a revised version of the manuscript that addresses the points raised during the review process. Specifically, both reviewers continue to have substantial concerns in a number of areas. I concur with those concerns. Please pay careful attention to each of the comments.

We look forward to receiving your revised manuscript.

Kind regards,

Yu-Wei Ryan Chen, PhD

Academic Editor

PLOS One

Journal Requirements:

Reviewers' comments:

Reviewer's Responses to Questions

**Comments to the Author**

1. If the authors have adequately addressed your comments raised in a previous round of review and you feel that this manuscript is now acceptable for publication, you may indicate that here to bypass the “Comments to the Author” section, enter your conflict of interest statement in the “Confidential to Editor” section, and submit your "Accept" recommendation.

Reviewer #1: (No Response)

Reviewer #2: (No Response)

2. Is the manuscript technically sound, and do the data support the conclusions?

Reviewer #1: Partly

Reviewer #2: Partly

3. Has the statistical analysis been performed appropriately and rigorously? 

Reviewer #1: Yes

Reviewer #2: Yes

4. Have the authors made all data underlying the findings in their manuscript fully available?

Reviewer #1: Yes

Reviewer #2: Yes

5. Is the manuscript presented in an intelligible fashion and written in standard English?

Reviewer #1: Yes

Reviewer #2: No

6. Review Comments to the Author

Reviewer #1: This is a thoughtful, and much-needed contribution to the literature on adult DCD. With modest refinements the paper will be well positioned to inform research, practice, and policy, while remaining true to the lived experiences at its core. The findings are shared with care, reflexivity, and ethical sensitivity. Participant quotations are clear and emotionally resonant, effectively conveying experiences of frustration, invalidation, and systemic neglect. The thematic structure is now clearer and more reader-friendly and the discussion of societal awareness and systemic abandonment is a strong addition.

However, there are still some areas that need attention before ready for publication.

At present, the manuscript sometimes implies that DCD is uniquely neglected compared to other neurodevelopmental conditions. Whilst I agree with this, it would be good to unpack which systems (healthcare, education, employment, community) are most inequitable, and whether this reflects absolute neglect or relative invisibility compared to conditions such as ADHD or ASD.

Additionally, as approximately half of the participants reported co-occurring neurodevelopmental conditions, the analysis and interpretation at times appears to attribute outcomes such as cognitive load, concentration difficulties, fatigue, and executive functioning challenges primarily to DCD. Greater interpretive caution is required in these areas and it would be helpful to more explicitly distinguish between participant-reported attributions and condition-specific effects, and to integrate consideration of co-occurring conditions more consistently throughout the Results and Discussion sections rather than primarily within the limitations. This would strengthen the validity and transparency of the conclusions.

The paper is framed as taking a lifespan and cross-cultural approach. However, the empirical basis for cross-cultural analysis is not justified, as the sample is predominantly UK-based, with only one participant residing overseas, the study cannot reasonably be framed as international, and this wording should be amended throughout.

Similarly, while participants retrospectively reflect on childhood and adulthood experiences, the paper would benefit from greater clarity regarding how a “lifespan approach” is operationalised analytically. Explicitly articulating how experiences across different developmental stages were examined would improve conceptual clarity.

Details

Abstract

Even though you do not have the ages of the ppts, I would tell the reader their sex in the abstract

Intro

The opening section needs to more clearly establish that DCD is fundamentally a movement condition; this is not sufficiently explicit in the initial description. Although the four DSM-5 criteria are listed, this approach is overly basic and reads at an undergraduate level rather than reflecting the conceptual depth expected in a published journal article. A more sophisticated, narrative description of the condition is needed.

The introduction should be written in tight APA style, with only essential information included. Currently, key messages are diluted by unnecessary dialogue. For example, the sentence “This is also reflected in academic literature” is redundant and should be removed, with the relevant references cited directly.

While the manuscript states that DCD manifests in adulthood, greater detail is required to explain how this occurs. The paragraph would benefit from clearer elaboration of adult impacts, such as ongoing difficulties in education (e.g. handwriting and fine motor skills in higher education) and challenges with independent living skills.

References are frequently incorrectly positioned throughout the introduction:

• Citations should follow the evidence presented, not appear at the beginning of sentences.

• Several references are missing spaces between the citation and subsequent text.

• Phrases such as “Previous literature” should be followed by citations, not preceded by them. For example, sentences beginning with bracketed references (e.g. “[14] Previous literature…”, “[18] This study therefore aimed…”) should be revised for correctness and readability.

The discussion of prior qualitative research needs refinement. While the manuscript states that existing qualitative studies focus on narrow aspects of adulthood (e.g. university life or driving), the authors should check whether additional qualitative studies exist and ensure the claim is accurate and well-supported.

Methods

The statement that participants were recruited by contacting “all English-speaking national DCD charities globally” requires quantification and clarification. The authors should specify how many charities were contacted, In which countries, and avoid overstating the global reach of recruitment, as the sample does not reflect this claim.

The inclusion of a table listing interview questions may read more smoothly if these are integrated into prose, rather than presented in tabular form.

Procedure

When stating that “Participants were asked seven questions, with prompts for further discussion used where necessary”, please remind the reader where to locate these questions in the paper.

Results

Although the table of themes is informative, it currently attempts to do too much. Standard qualitative reporting practice would involve presenting a thematic map or list of themes only, and introducing example data later within the detailed analysis. Revising the table to function purely as a thematic overview would improve clarity and help readers better contextualise the findings.

That said, the results section now reads much more clearly, and the revised structure significantly improves coherence and accessibility.

Discussion

Claims that the findings apply across the lifespan are overstated. While the study includes reflections on childhood and adulthood, it only briefly addresses selected aspects of the lifespan and should be framed more cautiously.

In several places, references are cited without sufficient explanation or integration into the discussion. For example, the statement “similarly to previous research, motor skill challenges in childhood (subtheme 1.1) were seen to play a central role in the experiences of individuals with DCD [8]” is too abstract; the cited work needs to be more clearly explained and explicitly linked to the current findings.

There are repeated issues with the method of citation, where prior research is not properly embedded within the narrative. For example:

“[6,20][14] Previous research has alluded to individuals with DCD finding learning to drive challenging [14].”

This structure is awkward and does not meaningfully integrate the literature into the discussion.

Similarly, when discussing cognitive load and driving, the sentence referencing anxiety (“…which was anxiety inducing [36]”) lacks context. The reader needs to understand how reference [36] relates conceptually or empirically to the present findings.

The paragraph on inequitable support across neurodevelopmental conditions is conceptually strong but poorly referenced. The standalone list of citations ([51–53][64][70][51–53]) is unclear and must be explicitly linked to the preceding claims about awareness, validation, and lived experience.

In the section on physical activity, childhood social exclusion is well described; however, the discussion should also briefly acknowledge the health implications of reduced physical activity in adulthood, including increased risks related to obesity and cardiovascular health, which are concerns for adults with DCD.

Finally, ensure terminological accuracy and consistency throughout; the correct diagnostic label is DSM-5, not DSM-V.

Minor Points

• Some sections would benefit from tighter language to reduce repetition.

• Ensure consistent use of terminology when discussing inequity versus invisibility.

• A final proofread for flow and concision in the discussion would further strengthen readability.

• Some constructs (e.g., “cognitive load,” “executive functioning”) would benefit from clearer definition. Clarifying whether these terms are used descriptively (as participant experiences) or in reference to established cognitive constructs would improve precision.

• Age data were not collected, which limits interpretation of generational effects. While this is acknowledged as a limitation, its implications for interpreting retrospective accounts could be more explicitly discussed.

Reviewer #2: Thank you for your openness and for engaging with the previous review. While several comments have been addressed and clear efforts have been made to respond, a number of substantive issues remain that require further consideration.

1. Abstract

A. Methods - Minor Comment

“Participants were recruited both nationally and internationally Lived experience interviews focused on experiences in primary care, education, friendships, wellbeing, employment and romantic relationships.”

Needs a period/full stop between “internationally” and “lived”

B. Methods – Major Comment

Please provide some demographic information about participants – males, females, mean age, etc.

I understand from the response to the other reviewer that some demographic data were not collected and thus mean age and SD cannot be provided. While the absence of age data is acknowledged in the Limitations section, it will also benefit this manuscript to acknowledge this early on in the methods section. This is particularly important given the manuscript’s emphasis on lifespan and developmental transitions. As such, it would be important to signal this constraint earlier (e.g., in Methods or when interpreting age-related experiences), or to further temper claims that rely on age or generational differences throughout the manuscript.

For example – it can be briefly acknowledged in Methods or Participants that participant age was not systematically collected, and therefore age-related analyses or generational interpretations are limited.

C. Results – major comment

“Emergent themes highlighted a major lack of societal awareness in all life domains, and difficulty navigating health, education and workplace systems for support, resulting in relative abandonment and a lack of validation within their lived experience of DCD.”

Thank you for addressing the earlier comment around this statement. This statement would benefit from more clarity. Presently, it is not clear from this statement who is impacted or experiences the difficulty navigating these systems. Is it difficulty for those with DCD or society. My understanding was low societal awareness, but difficulties were experienced by individuals with DCD.

2. Introduction

A. Minor comments – several areas need proof-reading and editing for grammatical accuracy. Eg: a) “it is also known as and referred to [as] Dyspraxia”. b) After references 6-8 there should be a space between the period and the next sentence starting with “The”. c) “Despite between 30-70% of children [being] diagnosed with DCD”. d) “with studies highlighting that these challenges extended beyond the t motor skills”. e) with tasks such as cooking and cleaning, leaving adults”

B. Major comments – “It is, however, important to note that DCD is often diagnosed with a co-occurring condition, such as Autism (50%) and ADHD (30%) [16], more research is needed to disentangle the role DCD plays in these cognitive challenges.”

Thank you for attempting to clarify that executive functioning challenges in DCD could be impacted by DCD but also by co-occurring NDC. Unfortunately, this statement is not very clear and will benefit from restructuring. Secondly, I’m wondering if this is an appropriate place for this statement. if the manuscript is not intending to resolve this question of disentangling the role of DCD in these cognitive challenges, it should likely not be stated in this way in the introduction. It would be enough to acknowledge that DCD is well-known to co-occur with other neurodevelopmental conditions, eg, ASD, ADHD. Thirdly, and importantly, sometimes there are multiple co-occurring conditions alongside DCD, not just one other co-occurring condition. So, the language here should be clearer.

D. Additional major comment – analytic rationale

Please consider strengthening the rationale for the study in the latter part of the introduction. While the importance of DCD across the lifespan is discussed, the specific analytic contribution of the current qualitative study (beyond documenting lived experience) could be articulated more clearly. For example, clarifying what gaps in the existing qualitative literature this study addresses, and how it advances understanding beyond previous work, would strengthen the framing of the paper.

3. Methods

A. “The sample comprised 8 females and 2 males and 5 had a co-occurrence (two with Autism, two with Dyslexia, and one with ADHD).”

It would be good to include where/how these co-occurring diagnoses made, or who made them and at what age if this information is available. Was any evidence of these diagnoses assessed or gathered by the researchers? If not, then this should be acknowledged in the methods section. Such as, self-reported diagnoses, not verified independently by researchers, etc.

B. Procedure – “Interviews lasted between 30 minutes to an hour.” In the abstract it states 30-45 minutes. Please consider consistency with the information throughout the manuscript.

C. Ages – As noted above, I understand from the response to the other reviewer that demographic data were not collected and thus mean age and SD cannot be provided. Please acknowledge this constraint explicitly in this section. Doing so only in the limitations section may not be adequate. For example – it can be briefly acknowledged in Methods or Participants that participant age was not systematically collected, and therefore age-related analyses or generational interpretations are limited.

4. Results

A. “The current analysis presents the difficult road travelled by adults with DCD both contemporaneously and retrospectively. The data transcends symptomology to wider system issues, including lack of societal awareness and effective support which impact the lived experience of health and wellbeing within the DCD diagnosis. Stories of abandonment, struggles to be heard, and battles to access effective support across sectors, collectively revealed a society which is failing to scaffold vulnerable adults, and by extension, children”

This paragraph would be more appropriate in the discussion section as it draws all of the themes together and is more conclusive. The results section should ideally be only stating the results of the study, i.e., the themes and sub-themes that came about from the raw data as has been done wonderfully in the remainder of the section.

B. Theme 1 - “in rites of passage”. It may be more appropriate to state this as lifecycle transitions or stereotypical/cultural developmental stages.

C. Subtheme 1.2: “Belittling”

C.1. It may not be appropriate to use emotive terms e.g. “belittling” unless that was what was specifically reported by participants. Eg; One participant reported “I felt belittled”. It is preferable to use neutral language when summarising both in the heading and in-text where the word belittling has been applied. Example of appropriate use – “some participants drew upon the label ‘clumsy’ on reflecting upon their experiences in childhood”.

C.2. “However, the reality of their ‘clumsiness’ sometimes resulted in direct implications for their health and wellbeing, as well as scrutiny for their family.”

It may be helpful to more clearly distinguish between participants’ self-descriptions and the authors’ analytic framing. Additionally, statements referring to “direct implications for health and wellbeing” and “scrutiny for their family” are somewhat broad and would benefit from clarification or more precise description of the nature and source of these impacts. Adopting more neutral, descriptive language and specifying the mechanisms or contexts involved may strengthen clarity and interpretability.

D. Subtheme – “rites of passage”

Please consider using more appropriate terminology such as functional milestones, developmental stages, lifecycle transitions, etc than rites of passage.

The discussion of driving-related difficulties would benefit from more cautious interpretation. While the authors appropriately note the presence of co-occurring ADHD in one participant, the subsequent attribution of driving challenges to cognitive demands (e.g., attention, navigation, spatial awareness) may extend beyond what can be supported by the qualitative data presented. It may be helpful to clarify that these interpretations reflect participants’ reported experiences rather than inferred mechanisms, and to avoid implying that cognitive difficulties are independent of, or separable from, motor coordination challenges. Additionally, greater caution may be warranted when generalising across participants with and without co-occurring conditions.

E. Subtheme Cognitive Load

“The cognitive load associated with driving was also seen as problematic in relation to planning and processing more broadly. Participants revealed the reality of life within which taken for granted practices demanded significant efforts.”

“Additionally, the cognitive challenges participants with DCD experienced, sometimes left them feeling exhausted, even after ‘simple’ tasks.”

Please consider using more cautious interpretation and language than conclusive/causal language. Eg: participant-reported cognitive challenges.

F. Subtheme Commonality in difference in adult friendships

The language in this section is relatively abstract and interpretive. Greater specificity regarding participants’ reported experiences of peer relationships, and clearer explanation of terms such as “commonality in difference,” would improve clarity and readability.

G. Subtheme 1.7: Commonality in difference in romantic relationships

Please consider correcting the formatting between the paragraph and the heading. Please also consider explaining “commonality in difference” in this section as well to clarify meaning.

H. Subtheme 1.8: Lack of equity in employment opportunities

The interpretive framing may benefit from greater caution and clarity. In particular, while participants often attributed workplace challenges to DCD, a substantial proportion of the quoted examples involve individuals with co-occurring neurodevelopmental conditions (e.g., ADHD, Autism, Dyslexia). Given this, it may be helpful to more explicitly acknowledge the potential influence of co-occurring conditions and to avoid attributing these experiences primarily to DCD alone. Clarifying that these interpretations reflect participants’ perceptions, rather than definitive causal attribution, would strengthen the analytic transparency. Additionally, the reference to “lack of equity in employment opportunities” could be further specified or supported, as it currently reads as a broad conclusion relative to the qualitative evidence presented.

I. “Participants often felt that there was a lack of interest and investment in DCD at a societal level, which impacted them on an individual level, leaving some frustrated and tired from continually advocating for themselves.”

This would be stronger if framed in terms of participant-reports rather than “participants felt” for a less emotive, more neutral framing.

J. Subtheme 2.8: Inherent lack of awareness compared to other neurodevelopmental conditions

Both paragraphs would benefit from greater clarity around who experiences the lack of awareness is, and where the lack of awareness is observed. As it currently stands, the statements are vague.

Eg: “All participants expressed a profound lack of awareness of DCD, especially when compared to other NDCs.”

K. Subtheme 2.9: Huge hole in community level support and advocacy

Please consider reframing “huge hole” which comes across as colloquial. Consider “gap” or “considerable gap” instead.

“participants felt a sense of abandonment” please consider instead “participants reported a sense…”

L. Participant voice vs interpretation

Across several subthemes, there are instances where analytic language appears to move beyond participant descriptions into broader interpretation (e.g., societal failure, abandonment, inequity). Please consider more consistently signalling when statements reflect participant-reported experiences versus author interpretation. Explicitly distinguishing between these may strengthen analytic clarity.

M. Co-occurring conditions across themes

Please consider more consistently acknowledging the presence of co-occurring neurodevelopmental conditions across subthemes, rather than addressing this only in selected sections. Given the prominence of co-occurrence within the sample, a brief reminder within relevant subthemes may help avoid over-attribution of experiences solely to DCD.

5. Discussion

A. Please consider reframing emotive language to more neutral language, eg: “belittling”.

B. Please consider reframing “alluded” throughout the transcript to “reported”, “recounted”, “identified” etc.

C. Please consider reframing “rites of passage” to “functional milestones”, “developmental stages”, etc.

D. Please consider throughout the discussion section including existing qualitative research where similar themes have or have not been considered or addressed previously, and whether these themes are upheld or challenged relative to previous research, even if lifespan or childhood research. If none is available – this can also be acknowledged in the discussion section.

E. “appeared to suffer at the hands”. Please consider using neutral language rather than emotive language.

F. Causal and diagnostic implications

Please consider reviewing the discussion section holistically to ensure that language does not imply causal relationships or diagnostic utility that cannot be supported by the qualitative design. In several places, emotional experiences and functional challenges are discussed in ways that suggest DCD as a primary explanatory mechanism. Greater caution in attributing causality, and clearer framing of findings as participant-reported associations and interpretations, would strengthen the discussion.

G. Diagnostic relevance

Please consider tempering statements that suggest findings may directly support identification or diagnostic processes for DCD in adulthood. While the experiences described are clearly important, stronger empirical or theoretical justification would be required to support claims relating to diagnostic utility.

6. Limitations

A. co-morbid Autism – please consider using the term co-occurring.

B. “Although this reflects the reality of DCD diagnoses [16] in places it was challenging to disentangle causation, resulting in an element of speculation.”

Please reconsider this statement, a qualitative study cannot comment on causation regardless. This sentence would benefit from greater clarity and precision. The phrasing is somewhat vague and may understate an important methodological limitation. It may be helpful to more explicitly acknowledge the limits of causal interpretation inherent in the study design and to clarify where and how interpretive uncertainty arose. Please also state how future research could address this and in what way.

While this issue is acknowledged in the limitations, similar causal or attributional language appears elsewhere in the manuscript, and addressing this solely within the limitations section may not be sufficient.

C. “and we were interviewing adults who may not have been able to access such historic evidence.” This is speculative as the researchers do not know whether or not the participants had access to this information.

D. Scope of inference

Please consider explicitly stating that the findings reflect the experiences of a small qualitative sample and may not be representative of all adults with DCD. Clarifying the scope and transferability of the findings would further strengthen this section.

7. Conclusion

A. “Individuals with DCD often feel alone, isolated and misunderstood” Please consider reframing to “often report feeling…”

B. “Data revealed that there is an urgent need for the profile of DCD to be raised” Before this statement please summarise the actual results of the data, i.e., data revealed themes and subthemes around different aspects of life, for example, … followed by the implication that these data indicate a need to raise the profile.

Unfortunately, the conclusion section requires more specificity as it is currently somewhat vague about the results and conclusions.

C. Alignment with data

Please consider ensuring that all concluding statements are clearly traceable to the themes and subthemes presented in the Results section. Where broader implications are drawn, it may be helpful to explicitly signal these as interpretations or implications rather than direct findings.

Overarching comment

While specific comments are provided throughout, my overarching comments and concerns relate to analytic framing across the manuscript, particularly the distinction between participant-reported experiences and author interpretation, the handling of co-occurring neurodevelopmental conditions, and the avoidance of causal or diagnostic implications. Addressing these issues holistically would substantially strengthen the manuscript.

7. PLOS authors have the option to publish the peer review history of their article (what does this mean?). If published, this will include your full peer review and any attached files.

Reviewer #1: No

Reviewer #2: No

---

## [Author Response · Author response to Decision Letter 2]

4 Mar 2026

Response to reviewers

Reviewer 1

This is a thoughtful, and much-needed contribution to the literature on adult DCD. With modest refinements the paper will be well positioned to inform research, practice, and policy, while remaining true to the lived experiences at its core. The findings are shared with care, reflexivity, and ethical sensitivity. Participant quotations are clear and emotionally resonant, effectively conveying experiences of frustration, invalidation, and systemic neglect. The thematic structure is now clearer and more reader-friendly and the discussion of societal awareness and systemic abandonment is a strong addition.

We are sincerely grateful to the reviewer for their generous and thoughtful feedback. It is encouraging to know that the revisions made in the previous round have strengthened the manuscript in these important areas, and we have endeavoured to build upon this foundation in the current revision.

However, there are still some areas that need attention before ready for publication.

At present, the manuscript sometimes implies that DCD is uniquely neglected compared to other neurodevelopmental conditions. Whilst I agree with this, it would be good to unpack which systems (healthcare, education, employment, community) are most inequitable, and whether this reflects absolute neglect or relative invisibility compared to conditions such as ADHD or ASD.

We appreciate the reviewer raising this nuanced and important point, and we agree that unpacking systemic inequity in greater depth would be a valuable contribution. However, we respectfully note that the qualitative design and data collected in the current study do not allow for systematic comparison across sectors in terms of the relative degree of inequity experienced. Whilst the results section does address the lack of awareness and support across healthcare, education, and the wider community, we feel that drawing comparisons between sectors would risk extending beyond what the data can meaningfully support. In the discussion, we have referenced existing literature that speaks to differential levels of awareness and support across neurodevelopmental conditions, and we feel this provides appropriate contextualisation without overstating the interpretive reach of the current findings.

Additionally, as approximately half of the participants reported co-occurring neurodevelopmental conditions, the analysis and interpretation at times appears to attribute outcomes such as cognitive load, concentration difficulties, fatigue, and executive functioning challenges primarily to DCD. Greater interpretive caution is required in these areas and it would be helpful to more explicitly distinguish between participant-reported attributions and condition-specific effects, and to integrate consideration of co-occurring conditions more consistently throughout the Results and Discussion sections rather than primarily within the limitations. This would strengthen the validity and transparency of the conclusions.

We agree that greater and more consistent interpretive caution was warranted throughout the manuscript. In response, we have made a concerted effort to revise the language across both the Results and Discussion sections to ensure that all data is clearly framed as reflecting participant-reported experiences rather than researcher-inferred causal mechanisms. We feel it is important to present participants' own attributions of their challenges to DCD, particularly given that cognition was not included as an interview prompt and yet emerged organically and consistently across all participant accounts - a finding which we feel is itself noteworthy and worthy of discussion. Nonetheless, we fully appreciate the need for analytic transparency regarding co-occurring conditions, and have therefore included explicit acknowledgement of the potential role of co-occurring diagnoses within each relevant subtheme in the Results section, and have woven this consideration throughout the Discussion to ensure the reader is consistently equipped to interpret the findings with appropriate caution.

The paper is framed as taking a lifespan and cross-cultural approach. However, the empirical basis for cross-cultural analysis is not justified, as the sample is predominantly UK-based, with only one participant residing overseas, the study cannot reasonably be framed as international, and this wording should be amended throughout. Similarly, while participants retrospectively reflect on childhood and adulthood experiences, the paper would benefit from greater clarity regarding how a “lifespan approach” is operationalised analytically. Explicitly articulating how experiences across different developmental stages were examined would improve conceptual clarity.

Upon reflection, we agree that framing the study as adopting a lifespan or cross-cultural approach was not adequately supported by the empirical basis of the sample. As a result, we have removed all references to a lifespan approach throughout the manuscript. The sole reference to cross-cultural considerations that remains appears in the Limitations section, where we acknowledge that cross-cultural comparison was not possible given that only one participant was based internationally - a limitation that was flagged by previous reviewers and which we felt was important to retain for transparency.

Abstract

Even though you do not have the ages of the ppts, I would tell the reader their sex in the abstract

We thank the reviewer for this suggestion and agree that including participant sex in the abstract provides useful contextual information for the reader. We have now added this information to the abstract accordingly.

Intro

The opening section needs to more clearly establish that DCD is fundamentally a movement condition; this is not sufficiently explicit in the initial description. Although the four DSM-5 criteria are listed, this approach is overly basic and reads at an undergraduate level rather than reflecting the conceptual depth expected in a published journal article. A more sophisticated, narrative description of the condition is needed.

This section was modified to include the DSM-5 criteria in response to the previous round of revisions, however we have now revised it further to incorporate both a nuanced narrative description of DCD as a fundamentally movement-based condition and a more conceptually integrated presentation of the DSM-5 criteria.

The introduction should be written in tight APA style, with only essential information included. Currently, key messages are diluted by unnecessary dialogue. For example, the sentence “This is also reflected in academic literature” is redundant and should be removed, with the relevant references cited directly.

While the manuscript states that DCD manifests in adulthood, greater detail is required to explain how this occurs. The paragraph would benefit from clearer elaboration of adult impacts, such as ongoing difficulties in education (e.g. handwriting and fine motor skills in higher education) and challenges with independent living skills.

We have carefully reviewed the introduction and removed unnecessary dialogue to ensure the writing is concise and appropriately focused. We have also expanded the discussion of how DCD manifests in adulthood, including greater elaboration on the impact of ongoing motor difficulties in higher education and on independent living skills, ensuring that the adult experience of DCD is more clearly and substantively conveyed from the outset of the paper.

References are frequently incorrectly positioned throughout the introduction:

• Citations should follow the evidence presented, not appear at the beginning of sentences.

• Several references are missing spaces between the citation and subsequent text.

• Phrases such as “Previous literature” should be followed by citations, not preceded by them. For example, sentences beginning with bracketed references (e.g. “[14] Previous literature…”, “[18] This study therefore aimed…”) should be revised for correctness and readability.

There are repeated issues with the method of citation, where prior research is not properly embedded within the narrative. For example:

“[6,20][14] Previous research has alluded to individuals with DCD finding learning to drive challenging [14].”This structure is awkward and does not meaningfully integrate the literature into the discussion.

The paragraph on inequitable support across neurodevelopmental conditions is conceptually strong but poorly referenced. The standalone list of citations ([51–53][64][70][51–53]) is unclear and must be explicitly linked to the preceding claims about awareness, validation, and lived experience.

We apologise for these formatting inconsistencies, which arose from technical issues with the reference management software used during manuscript preparation. We have now thoroughly reviewed and corrected all in-text citations throughout the manuscript to ensure they are consistently and accurately positioned.

The discussion of prior qualitative research needs refinement. While the manuscript states that existing qualitative studies focus on narrow aspects of adulthood (e.g. university life or driving), the authors should check whether additional qualitative studies exist and ensure the claim is accurate and well-supported.

We thank the reviewer for prompting us to revisit this claim. In response, we conducted an updated literature review and identified one additional qualitative study published after the original submission of this manuscript. This paper has now been incorporated into the introduction, and we have taken care to more clearly articulate the specific gaps in the existing qualitative literature that the current study addresses.

Methods

The statement that participants were recruited by contacting “all English-speaking national DCD charities globally” requires quantification and clarification. The authors should specify how many charities were contacted, In which countries, and avoid overstating the global reach of recruitment, as the sample does not reflect this claim.

We have amended the manuscript to specify that four national DCD charities were contacted, along with the continents in which they are based. We have refrained from specifying individual countries, as doing so could risk compromising participant anonymity given the limited DCD infrastructure in some regions.

The inclusion of a table listing interview questions may read more smoothly if these are integrated into prose, rather than presented in tabular form.

We were asked by previous reviewers to include this information in tabular form. As writing these in prose will not fundamentally change the content, we have decided to keep the table in the manuscript.

Procedure

When stating that “Participants were asked seven questions, with prompts for further discussion used where necessary”, please remind the reader where to locate these questions in the paper.

We have now added (see Table 1) to the manuscript for ease.

Results

Although the table of themes is informative, it currently attempts to do too much. Standard qualitative reporting practice would involve presenting a thematic map or list of themes only, and introducing example data later within the detailed analysis. Revising the table to function purely as a thematic overview would improve clarity and help readers better contextualise the findings.

We have now removed the example data from this table for ease of interpretation.

Discussion

Claims that the findings apply across the lifespan are overstated. While the study includes reflections on childhood and adulthood, it only briefly addresses selected aspects of the lifespan and should be framed more cautiously.

We agree and, as noted above, have removed all references to a lifespan approach throughout the manuscript.

In several places, references are cited without sufficient explanation or integration into the discussion. For example, the statement “similarly to previous research, motor skill challenges in childhood (subtheme 1.1) were seen to play a central role in the experiences of individuals with DCD [8]” is too abstract; the cited work needs to be more clearly explained and explicitly linked to the current findings.

Similarly, when discussing cognitive load and driving, the sentence referencing anxiety (“…which was anxiety inducing [36]”) lacks context. The reader needs to understand how reference [36] relates conceptually or empirically to the present findings.

We have been through the manuscript to ensure all cited literature is explained in relation to the current study.

In the section on physical activity, childhood social exclusion is well described; however, the discussion should also briefly acknowledge the health implications of reduced physical activity in adulthood, including increased risks related to obesity and cardiovascular health, which are concerns for adults with DCD.

We are grateful to the reviewer for raising this important point, which adds meaningful depth to the discussion of physical activity in the context of DCD. We have now incorporated a sentence acknowledging the health implications of reduced physical activity in adulthood, including the increased risk of obesity and cardiovascular disease, accompanied by appropriate supporting references.

Finally, ensure terminological accuracy and consistency throughout; the correct diagnostic label is DSM-5, not DSM-V.

We have now updated this in the manuscript and ensure all acronyms are accurate throughout.

Minor Points

• Some sections would benefit from tighter language to reduce repetition.

• Ensure consistent use of terminology when discussing inequity versus invisibility.

• A final proofread for flow and concision in the discussion would further strengthen readability.

We have carefully proofread the manuscript in its entirety, addressing areas of repetition, improving overall flow, and ensuring terminological consistency throughout.

• Some constructs (e.g., “cognitive load,” “executive functioning”) would benefit from clearer definition. Clarifying whether these terms are used descriptively (as participant experiences) or in reference to established cognitive constructs would improve precision.

We have revised the relevant sections of both the Results and Discussion to clearly signal when these terms are being used descriptively to reflect participant-reported experiences, and when they are being used in reference to established theoretical or cognitive constructs.

• Age data were not collected, which limits interpretation of generational effects. While this is acknowledged as a limitation, its implications for interpreting retrospective accounts could be more explicitly discussed.

We agree with the reviewer that the absence of age data warranted more substantive discussion than was provided in the original manuscript. We have therefore expanded the relevant section in the Limitations to more explicitly address how the lack of age data constrains our ability to situate participants' retrospective accounts within specific historical or generational contexts, to identify whether variations in experience reflect generational differences or individual variation, and to determine how the passage of time between participants' experiences and the interview may have affected both memory accuracy and narrative framing. We have also noted that future research would benefit from the collection of age data to enable more nuanced cohort-level interpretation.

Reviewer 2

Thank you for your openness and for engaging with the previous review. While several comments have been addressed and clear efforts have been made to respond, a number of substantive issues remain that require further consideration.

“Participants were recruited both nationally and internationally Lived experience interviews focused on experiences in primary care, education, friendships, wellbeing, employment and romantic relationships.” Needs a period/full stop between “internationally” and “lived”

Thank you for highlighting this omission. We have included this and proofread the paper.

B. Methods – Major Comment

Please provide some demographic information about participants – males, females, m

---

## [Decision Letter · Decision Letter 2]

28 Apr 2026

PONE-D-25-33989R2‘The world is just so fast, and I’m not fast… it’s just really, really difficult to keep up’: A qualitative exploration of the lived experience of adults with Developmental Coordination DisorderPLOS One

Dear Dr. Murray,

Thank you for submitting your manuscript to PLOS ONE. After careful consideration, we feel that it has merit but does not fully meet PLOS ONE’s publication criteria as it currently stands. Both reviewers have provided some minor suggestions for improvement. Therefore, we invite you to submit a revised version of the manuscript that addresses the points raised during the review process.

We look forward to receiving your revised manuscript.

Kind regards,

Yu-Wei Ryan Chen, PhD

Academic Editor

PLOS One

Journal Requirements:

Reviewers' comments:

Reviewer's Responses to Questions

**Comments to the Author**

1. If the authors have adequately addressed your comments raised in a previous round of review and you feel that this manuscript is now acceptable for publication, you may indicate that here to bypass the “Comments to the Author” section, enter your conflict of interest statement in the “Confidential to Editor” section, and submit your "Accept" recommendation.

Reviewer #1: All comments have been addressed

Reviewer #2: All comments have been addressed

2. Is the manuscript technically sound, and do the data support the conclusions?

Reviewer #1: Yes

Reviewer #2: Yes

3. Has the statistical analysis been performed appropriately and rigorously? 

Reviewer #1: Yes

Reviewer #2: Yes

4. Have the authors made all data underlying the findings in their manuscript fully available?

Reviewer #1: Yes

Reviewer #2: Yes

5. Is the manuscript presented in an intelligible fashion and written in standard English?

Reviewer #1: Yes

Reviewer #2: Yes

6. Review Comments to the Author

Reviewer #1: This manuscript addresses an important and underexplored area, offering valuable insight into the lived experiences of adults with DCD. The qualitative data presented are rich and meaningful, and the paper makes a worthwhile contribution to the literature. While the manuscript is generally well-structured and clearly written, a small number of revisions would strengthen clarity and impact. Only a few minor queries remain outstanding, and I would recommend publication once these have been addressed.

Introduction

The statement that tasks such as cooking and cleaning leave adults with DCD feeling “tired” [12] may not fully convey the extent of fatigue and overwhelm described in the literature and by lived experience. Consider strengthening this wording to better reflect the intensity and cumulative burden of these everyday activities for adults with DCD.

Results

Some sub-themes include a substantial amount of dialogue-based evidence, which is rich and insightful. However, this section would benefit from greater concision, as its current length may risk reducing reader engagement. Streamlining the number of quotations; selecting two to three representative examples per sub-theme would enhance clarity and reinforce the key messages without diminishing the depth of analysis.

Reviewer #2: Overall Recommendation: Minor Revision

The authors have engaged constructively with prior feedback and the manuscript has improved substantially. The study makes a valuable contribution to the qualitative literature on adults with DCD.

One minor point remains regarding the Discussion: in a few places, participant-reported experiences are still framed in ways that extend toward broader societal-level conclusions (e.g., inequity, systemic lack of support). While these interpretations are plausible, slightly more cautious wording would better align with the qualitative scope of the data.

These are relatively minor refinements, and I have no further major concerns.

7. PLOS authors have the option to publish the peer review history of their article (what does this mean?). If published, this will include your full peer review and any attached files.

Reviewer #1: **Yes:** Judith Marguerite Gentle

Reviewer #2: No

---

## [Author Response · Author response to Decision Letter 3]

6 May 2026

Reviewer #1: This manuscript addresses an important and underexplored area, offering valuable insight into the lived experiences of adults with DCD. The qualitative data presented are rich and meaningful, and the paper makes a worthwhile contribution to the literature. While the manuscript is generally well-structured and clearly written, a small number of revisions would strengthen clarity and impact. Only a few minor queries remain outstanding, and I would recommend publication once these have been addressed.

We thank the reviewer for their kind words, and for their support in getting the manuscript to this point!

Introduction

The statement that tasks such as cooking and cleaning leave adults with DCD feeling “tired” [12] may not fully convey the extent of fatigue and overwhelm described in the literature and by lived experience. Consider strengthening this wording to better reflect the intensity and cumulative burden of these everyday activities for adults with DCD.

Thank you for the suggestion! This had been toned down based on previous reviewer feedback, however, we agree that the previous wording does not fully capture experiences of those with DCD so we have changed this to: ‘enduring the cumulative burden of consequential fatigue which plays a significant role in their day to day lives’.

Results

Some sub-themes include a substantial amount of dialogue-based evidence, which is rich and insightful. However, this section would benefit from greater concision, as its current length may risk reducing reader engagement. Streamlining the number of quotations; selecting two to three representative examples per sub-theme would enhance clarity and reinforce the key messages without diminishing the depth of analysis.

We thank the reviewer for highlighting the length of the manuscript may impact reader engagement. As a response, we have been through the quotes in the manuscript and reduced all sections to a maximum of 3 quotes.

Reviewer #2: Overall Recommendation: Minor Revision

The authors have engaged constructively with prior feedback and the manuscript has improved substantially. The study makes a valuable contribution to the qualitative literature on adults with DCD.

We thank the reviewer acknowledging the work undertaken to the manuscript to get to this point. We really appreciate their support thus far.

One minor point remains regarding the Discussion: in a few places, participant-reported experiences are still framed in ways that extend toward broader societal-level conclusions (e.g., inequity, systemic lack of support). While these interpretations are plausible, slightly more cautious wording would better align with the qualitative scope of the data.

We thank the reviewer for this observation. We recognise that in places the Discussion extended beyond the scope of the qualitative data by framing participant experiences as broader societal-level conclusions. We have revisited the Discussion and adjusted the language throughout to ensure that claims are appropriately anchored to participants' accounts. For example, 'collectively revealed a society which is failing to scaffold vulnerable adults' has been revised to foreground participant voice, and phrases such as 'inequitable support' and 'reveals the need' have been reframed as participant perceptions rather than stated as objective fact. We believe these changes better reflect the qualitative scope of the data whilst preserving the substance of the findings.

---

## [Editor Report · Decision Letter 3]

13 May 2026

‘The world is just so fast, and I’m not fast… it’s just really, really difficult to keep up’: A qualitative exploration of the lived experience of adults with Developmental Coordination Disorder

PONE-D-25-33989R3

Dear Dr. Murray,

We’re pleased to inform you that your manuscript has been judged scientifically suitable for publication and will be formally accepted for publication once it meets all outstanding technical requirements.

Kind regards,

Yu-Wei Ryan Chen, PhD

Academic Editor

PLOS One
---

## [Editor Report · Acceptance letter]

PONE-D-25-33989R3

PLOS One

Dear Dr. Murray,

I'm pleased to inform you that your manuscript has been deemed suitable for publication in PLOS One. Congratulations! Your manuscript is now being handed over to our production team.

Kind regards,

on behalf of

Dr. Yu-Wei Ryan Chen

Academic Editor

PLOS One